# Towards the Dynamics of a DNN Learning Symbolic Interactions

**Qihan Ren**[1*], **Junpeng Zhang**[1,2*], **Yang Xu**[3], **Yue Xin**[1], **Dongrui Liu**[1,4], **Quanshi Zhang**[1†]

[1]Shanghai Jiao Tong University  [2]Beijing Institute for General Artificial Intelligence
[3]Zhejiang University  [4]Shanghai Artificial Intelligence Laboratory
`{renqihan, zhangjp63, zqs1022}@sjtu.edu.cn`

## Abstract

This study proves the two-phase dynamics of a deep neural network (DNN) learning interactions. Despite the long disappointing view of the faithfulness of post-hoc explanation of a DNN, a series of theorems have been proven [27] in recent years to show that for a given input sample, a small set of interactions between input variables can be considered as primitive inference patterns that faithfully represent a DNN's detailed inference logic on that sample. Particularly, Zhang et al. [41] have observed that various DNNs all learn interactions of different complexities in two distinct phases, and this two-phase dynamics well explains how a DNN changes from under-fitting to over-fitting. Therefore, in this study, we mathematically prove the two-phase dynamics of interactions, providing a theoretical mechanism for how the generalization power of a DNN changes during the training process. Experiments show that our theory well predicts the real dynamics of interactions on different DNNs trained for various tasks.

## 1 Introduction

**Background: mathematically guaranteeing that the inference score of a DNN can be faithfully explained as symbolic interactions.** Explaining the detailed inference logic hidden behind the output score of a DNN is considered one of the core issues for the post-hoc explanation of a DNN. However, after a comprehensive survey of various explanation methods, many studies [28, 1, 12] have unanimously and empirically arrived at a disappointing view of the faithfulness of almost all post-hoc explanation methods. Fortunately, the recent progress [27] has mathematically proven that given a specific input sample $x = [x_1, \cdots, x_n]^\top$, a DNN[3] for a classification task usually only encodes a small set of interactions between input variables in the sample. It is proven that these interactions act like primitive inference patterns and can accurately predict all network outputs, no matter how we randomly mask the input sample[4]. An *interaction* refers to a non-linear relationship encoded by the DNN between a set of input variables in $S$. For example, as Figure 1 shows, a DNN may encode a non-linear relationship between the three image patches in $S = \{x_1, x_2, x_3\}$ to form a *dog-snout* pattern, which makes a numerical effect $I(S)$ on the network output. The *complexity* (or *order*) of an interaction is defined as the number of input variables in the set $S$, *i.e.*, $\mathrm{order}(S) \overset{\mathrm{def}}{=} |S|$.

**Our task.** Since Zhou et al. [44] found that high-order (complex) interactions usually have a much higher risk of over-fitting than low-order (simple) interactions, in this study, we hope to further track

---

[*]Equal contribution.

[†]Quanshi Zhang is the corresponding author. He is with the Department of Computer Science and Engineering, the John Hopcroft Center, at the Shanghai Jiao Tong University, China. `zqs1022@sjtu.edu.cn`.

[3]The proof in [27] requires the DNN to generate relatively stable inference outputs on masked samples, which is formulated by three mathematical conditions (see Appendix B). It is found that DNNs for image classification, 3D point cloud classification, tabular data classification, and text generation

[4]It is proven that no matter how we randomly mask variables of the input sample, we can always use numerical effects of a few interactions to accurately regress the network outputs on all masked samples.

38th Conference on Neural Information Processing Systems (NeurIPS 2024).

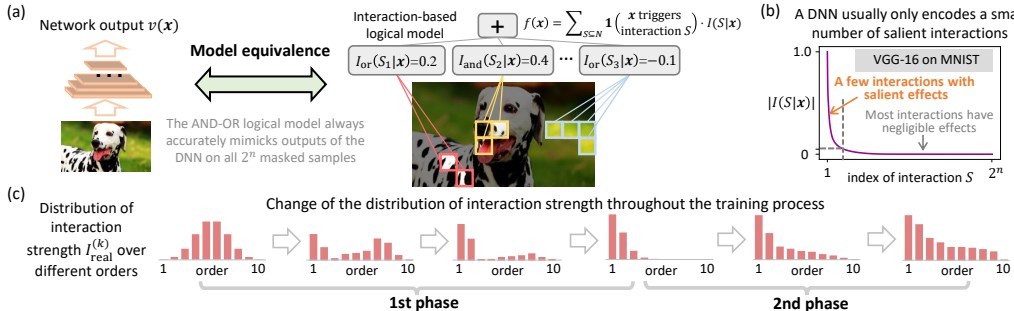

Figure 1: (a) It is proven that the DNN's inference on a certain sample is equivalent to a logical model that uses a small number of AND-OR interactions for inference. Each interaction corresponds to a non-linear (AND or OR) relationship between a set $S$ of input variables (*e.g.*, image patches). (b) Sparsity of interactions. We show the strength $|I(S|\boldsymbol{x})|$ of all $2^n$ interactions sorted in descending order. (c) Illustration of the two-phase dynamics of a DNN learning interactions of different orders.

the change in the complexity of interactions during training, so as to explain the change of the DNN's generalization power during training. In particular, the time when the DNN starts to learn high-order (complex) interactions indicates the starting point of over-fitting.

**Specifically, we focus on the two-phase dynamics of interaction complexity which was empirically observed by [41], and we aim to mathematically prove this dynamics.** First, before training, a DNN with randomly initialized parameters mainly encodes interactions of medium complexities. As Figure 2 shows, the distribution of interactions appears spindle-shaped. Then, in the first phase, the DNN eliminates interactions of medium and high complexities, thereby mainly encoding interactions of low complexity. In the second phase, the DNN gradually learns interactions of increasing complexities. We have conducted experiments to train DNNs with various architectures for different tasks. It shows that our theory can well predict the learning dynamics of interactions in real DNNs.

**The proven two-phase dynamics explain hidden factors that push the DNN from under-fitting to over-fitting.** (1) In the first phase, the DNN mainly removes noise interactions, (2) In the second phase, the DNN gradually learns more complex and non-generalizable interactions toward over-fitting.

## 2 Related work

**Long-standing disappointment on the faithfulness of existing post-hoc explanation of DNNs.** Many studies [30, 40, 29, 2, 15] have explained the inference score of a DNN, but how to mathematically formulate and guarantee the faithfulness of the explanation is still an open problem. For example, using an interpretable surrogate model to approximate the output of a DNN [3, 11, 35, 34] is a classic explanation technique. However, the good matching between the DNN's output and the surrogate model's output cannot fully guarantee that the two models use exactly the same inference patterns and/or use the same attention. Therefore, many studies [28, 12, 1] have unanimously and empirically arrived at a disappointing view of the faithfulness of current explanation methods. Rudin [28] pointed out that inaccurate post-hoc explanations of DNNs would be harmful to high-stakes applications. Ghassemi et al. [12] showed various failure cases of current explanation methods in the healthcare field and argued that using these methods to aid medical decisions was a false hope.

**New progress towards proving the faithfulness of symbolic explanation of a DNN.** Despite the disappointing view of post-hoc explanation methods, we have established a theory system of interactions within three years, which includes more than 30 papers, to quantify the symbolic concepts encoded by a DNN and explain the hidden factors that determine the generalization power and robustness of a DNN. We revisit this theory system as follows.

• *Proving interactions act as faithful primitives inference patterns encoded by the DNN.* Recent achievements in the theory system of interactions have provided a new perspective to formulate primitive inference patterns encoded by a DNN. We discovered [23] and proved [27] that a DNN's inference logic on a certain sample can be explained by only a small number of interactions. Furthermore, we discovered that salient interactions usually represented common inference patterns shared by different samples (sample-wise transferability of interactions) [21], and proposed a method to extract generalizable interactions shared by different DNNs (model-wise transferability of interactions) [4].

The above studies indicated that salient interactions could be considered primitive inference patterns encoded by a DNN, which served as the theoretical foundation of this study. Based on interactions, we also defined and learned the optimal baseline value for the Shapley value [25], and explained the encoding of different types of visual patterns in DNNs for image classification [5, 6].

• *Using interactions to explain the representation power of DNNs.* Our recent studies showed that interactions well explained the hidden factors that determine the adversarial robustness [24], adversarial transferability [37], and generalization power [44] of a DNN. We also discovered and proved the representation bottleneck of a DNN in encoding middle-complexity interactions [7]. In addition, we proved that compared to a standard DNN, a Bayesian neural network (BNN) tended to avoid encoding complex interactions [26], thus explaining the good adversarial robustness of BNNs. We discovered and explained the phenomenon that DNNs tended to learn simple interactions more easily than complex interactions [22]. We found that complex interactions were less generalizable than simple interactions [44], and further discovered the two-phase dynamics of a DNN learning interactions of different complexities [41]. To this end, this study aims to theoretically prove the discovery in [41] to better understand the two-phase dynamics of interactions.

• *Using interactions to unify the common mechanism of various empirical deep learning methods.* We proved that fourteen attribution methods could all be explained as a re-allocation of interaction effects [8]. We proved that twelve existing methods to improve adversarial transferability all shared the common utility of suppressing the interactions between adversarial perturbation units [42].

## 3 Dynamics of interactions

### 3.1 Preliminary: interactions

Let us consider a DNN $v$ and an input sample $\boldsymbol{x} = [x_1, \cdots, x_n]^\top$ with $n$ input variables indexed by $N = \{1, \cdots, n\}$. In different tasks, one can define different input variables, *e.g.*, each input variable may represent an image patch for image classification or a word/token for text classification. Let us consider a scalar output[5] of a DNN, denoted by $v(\boldsymbol{x}) \in \mathbb{R}$. Previous studies [4, 43] show that *the output score $v(\boldsymbol{x})$ can be decomposed into the sum of AND interactions and OR interactions.*

$$v(\boldsymbol{x}) = v(\boldsymbol{x}_\emptyset) + \sum\nolimits_{\emptyset \neq S \subseteq N} I_{\text{and}}(S|\boldsymbol{x}) + \sum\nolimits_{\emptyset \neq S \subseteq N} I_{\text{or}}(S|\boldsymbol{x}), \tag{1}$$

where the computation of $I_{\text{and}}(S|\boldsymbol{x})$ and $I_{\text{or}}(S|\boldsymbol{x})$ will be introduced later in Eq. (2).

**How to understand the physical meaning of AND-OR interactions.** Suppose that we are given an input sample $\boldsymbol{x}$. According to Theorem 2, a non-zero interaction effect $I_{\text{and}}(S|\boldsymbol{x})$ indicates that the entire function of the DNN must equivalently encode an AND relationship between input variables in the set $S \subseteq N$, although the DNN does not use an explicit neuron to model such an AND relationship. As Figure 1 shows, when the image patches in the set $S_2 = \{x_1 = nose, x_2 = tongue, x_3 = cheek\}$ are all present (*i.e.*, not masked), the three regions form a *dog-snout* pattern, and make a numerical effect $I_{\text{and}}(S_2|\boldsymbol{x})$ to push the output score $v(\boldsymbol{x})$ towards the dog category. Masking any image patch in $S_2$ will deactivate the AND interaction and remove $I_{\text{and}}(S_2|\boldsymbol{x})$ from $v(\boldsymbol{x})$. This will be shown by Theorem 2. Likewise, $I_{\text{or}}(S|\boldsymbol{x})$ can be considered as the numerical effect of the OR relationship encoded by the DNN between input variables in the set $S$. As Figure 1 shows, when one of the patches in $S_1 = \{x_4 = spotty\ region1, x_5 = spotty\ region2\}$ is present, a *speckles* pattern is used by the DNN to make a numerical effect $I_{\text{or}}(S_1|\boldsymbol{x})$ on the network output $v(\boldsymbol{x})$.

**Definition and computation.** Given a DNN and an input $\boldsymbol{x}$, the AND-OR interactions between each specific set of input variables $S \subseteq N(S \neq \emptyset)$ are computed as follows [4, 43].

$$I_{\text{and}}(S|\boldsymbol{x}) = \sum\nolimits_{T \subseteq S}(-1)^{|S|-|T|}v_{\text{and}}\left(\boldsymbol{x}_T\right), \quad I_{\text{or}}(S|\boldsymbol{x}) = -\sum\nolimits_{T \subseteq S}(-1)^{|S|-|T|}v_{\text{or}}\left(\boldsymbol{x}_{N \setminus T}\right), \tag{2}$$

where $\boldsymbol{x}_T$ denotes the sample in which input variables in $N \setminus T$ are masked[6], while input variables in $T$ are unchanged. The network output on each masked sample $v(\boldsymbol{x}_T), T \subseteq N$, is decomposed into two

---

[5]For example, one may set $v(\boldsymbol{x})$ as the loss value on sample $\boldsymbol{x}$. For a multi-category classification task, one usually either set $v(\boldsymbol{x})$ to be the output score for the ground-truth category before the softmax operation, or follow[7] to set $v(\boldsymbol{x}) = \log \frac{p(y^{\text{truth}}|\boldsymbol{x})}{1-p(y^{\text{truth}}|\boldsymbol{x})}$. See Table 1 for a summary of mathematical settings for interactions.

[6]The masked states of input variables are represented by specific baseline values $\boldsymbol{b} = [b_1, \cdots, b_n]^\top$ by following [41]. See Appendix G.3 for the detailed setting of baseline values.

components: (1) the component $v_{\text{and}}(\boldsymbol{x}_T) = 0.5v(\boldsymbol{x}_T) + \gamma_T$ that exclusively contains AND interactions, and (2) the component $v_{\text{or}}(\boldsymbol{x}_T) = 0.5v(\boldsymbol{x}_T) - \gamma_T$ that exclusively contains OR interactions, subject to $v(\boldsymbol{x}_T) = v_{\text{and}}(\boldsymbol{x}_T) + v_{\text{or}}(\boldsymbol{x}_T)$. Appendix F.1 shows that $v_{\text{and}}(\boldsymbol{x}_T) = v(\boldsymbol{x}_\emptyset) + \sum_{\emptyset \neq S' \subseteq T} I_{\text{and}}(S'|\boldsymbol{x})$ and $v_{\text{or}}(\boldsymbol{x}_T) = \sum_{S' \subseteq N : S' \cap T \neq \emptyset} I_{\text{or}}(S'|\boldsymbol{x})$. The sparsest AND-OR interactions are extracted by minimizing the following objective [20]: $\min_{\{\gamma_T\}} \sum_{S \subseteq N} |I_{\text{and}}(S|\boldsymbol{x})| + |I_{\text{or}}(S|\boldsymbol{x})|$. Please see Appendix C for details about the computation and Appendix D for mathematical support of the coefficient in Eq. (2).

**Salient interactions and noisy patterns.** Let us enumerate all $2^n$ combinations of variables $S \subseteq N$, and compute the interaction effects $I_{\text{and}}(S|\boldsymbol{x})$ and $I_{\text{or}}(S|\boldsymbol{x})$. We can identify a few *salient interactions* from all these interactions, *i.e.*, interactions whose absolute value exceeds a threshold ($|I_{\text{and}}(S|\boldsymbol{x})| \geq \tau$ or $|I_{\text{or}}(S|\boldsymbol{x})| \geq \tau$). Other interactions have small effects and are termed *noisy patterns*.

**Theorem 1** (Sparsity property, proven by [27], and discussed in Appendix B). *Given a DNN $v$ and an input sample $\boldsymbol{x}$ with $n$ input variables, let $\Omega \stackrel{\text{def}}{=} \{S \subseteq N : |I_{\text{and}}(S|\boldsymbol{x})| \geq \tau\}$ denote the set of salient AND interactions whose absolute value exceeds a threshold $\tau$. If the DNN can generate relatively stable inference outputs $v(\boldsymbol{x}_S)$ on masked samples[7], then the size of the set $|\Omega|$ has an upper bound of $\mathcal{O}(n^\xi / \tau)$, where $\xi$ is an intrinsic parameter for the smoothness of the network function $v(\cdot)$. Empirically, $\xi$ is usually within the range of [1.9,2.2].*

**Theorem 2** (Universal matching property, proven in [4] and Appendix F.1). *Given an input sample $\hat{\boldsymbol{x}}$, let us construct the following surrogate logical model $f(\cdot)$ to use AND-OR interactions for inference, which are extracted from the DNN $v(\cdot)$ on the sample $\hat{\boldsymbol{x}}$. Then, the output of the surrogate logical model $f(\cdot)$ can always match the output of the DNN $v(\cdot)$, no matter how the input sample is masked.*

$$\forall S \subseteq N, f(\hat{\boldsymbol{x}}_S) = v(\hat{\boldsymbol{x}}_S), \; f(\hat{\boldsymbol{x}}_S) = v(\hat{\boldsymbol{x}}_\emptyset) + \underbrace{\sum_{T \subseteq N} I_{\text{and}}(T|\hat{\boldsymbol{x}}) \cdot \mathbb{1}\left(\begin{smallmatrix}\hat{\boldsymbol{x}}_S \text{ triggers} \\ \text{AND relation } T\end{smallmatrix}\right)}_{v_{\text{and}}(\boldsymbol{x}_S)} + \underbrace{\sum_{T \subseteq N} I_{\text{or}}(T|\hat{\boldsymbol{x}}) \cdot \mathbb{1}\left(\begin{smallmatrix}\hat{\boldsymbol{x}}_S \text{ triggers} \\ \text{OR relation } T\end{smallmatrix}\right)}_{v_{\text{or}}(\boldsymbol{x}_S)} \quad (3)$$

$$= v(\boldsymbol{x} = \hat{\boldsymbol{x}}_\emptyset) + \sum_{\emptyset \neq T \subseteq S} I_{\text{and}}(T|\boldsymbol{x} = \hat{\boldsymbol{x}}) + \sum_{T \subseteq N : T \cap S \neq \emptyset} I_{\text{or}}(T|\boldsymbol{x} = \hat{\boldsymbol{x}}) \quad (4)$$

$$\approx v(\boldsymbol{x} = \hat{\boldsymbol{x}}_\emptyset) + \sum_{T \in \Omega_{\text{and}} : \emptyset \neq T \subseteq S} I_{\text{and}}(T|\boldsymbol{x} = \hat{\boldsymbol{x}}) + \sum_{T \in \Omega_{\text{or}} : T \cap S \neq \emptyset} I_{\text{or}}(T|\boldsymbol{x} = \hat{\boldsymbol{x}}), \quad (5)$$

*where $\Omega_{\text{and}}$ is the set of all salient AND interactions, and $\Omega_{\text{or}}$ is the set of all salient OR interactions.*

**What makes the interaction-based explanation faithful.** The following four properties guarantee that the inference score of a DNN can be faithfully explained by symbolic interactions.

• *Sparsity property.* The sparsity property means that a DNN for a classification task usually only encodes a small number of AND interactions with salient effects, *i.e.*, for most of all $2^n$ subsets of input variables $S \subseteq N$, $I_{\text{and}}(S|\boldsymbol{x})$ has almost zero interaction effect. Specifically, the sparsity property has been widely observed on various DNNs for different tasks [21], and it is also theoretically proven (see Theorem 1). The number of AND interactions whose absolute value exceeds the threshold $\tau$ ($|I_{\text{and}}(S|\boldsymbol{x})| \geq \tau$), is $\mathcal{O}(n^\xi / \tau)$, where $\xi$ is empirically within the range of $[1.9, 2.2]$. This indicates that the number of salient interactions is much less than $2^n$. Furthermore, the sparsity property also holds for OR interactions, because an OR interaction can be viewed as a special kind of AND interaction[8].

• *Universal matching property.* The universal matching property means that the output of the DNN on a masked sample $\boldsymbol{x}_S$ can be well matched by the sum of interaction effects, no matter how we randomly mask the sample and obtain $\boldsymbol{x}_S$. This property is proven in Theorem 2.

• *Transferability property.* The transferability property means that salient interactions extracted from one input sample can usually be extracted from other input samples as well. If so, these interactions are considered transferable across different samples. This property has been widely observed by [21] on various DNNs for different tasks.

---

[7]This is formulated by three mathematical conditions. (1) The DNN does not encode highly complex interactions. (2) Let us compute the average classification confidence when we mask different random sets of $k$ input variables (generating $\{\boldsymbol{x}_T : |T| = n - k\}$). Then, the average confidence monotonically decreases when more input variables are masked. (3) The decreasing speed of the average confidence is polynomial. See Appendix B for the detailed mathematical formulation.

[8]If we flip the masked state and the presence state of each input variable (*i.e.*, taking $b_i$ as the presence state of the $i$-th variable, while taking $x_i$ as the masked state), then OR interactions can be viewed as a special kind of AND interactions. See Appendix E for details.

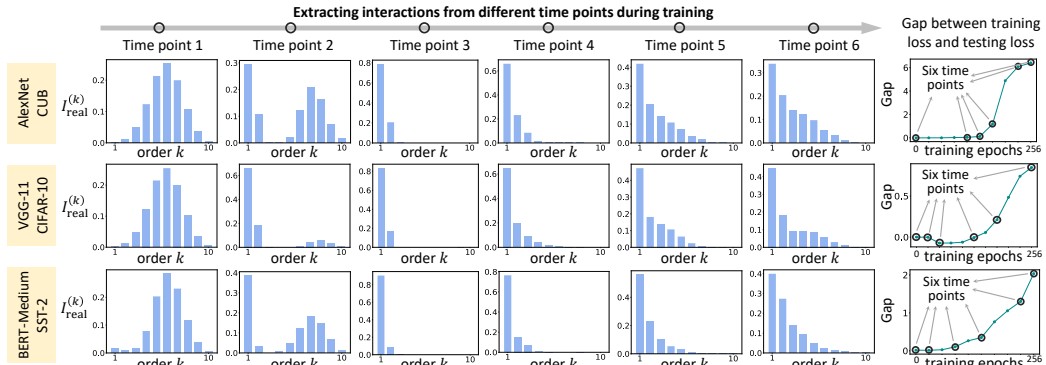

Figure 2: The distribution of interaction strength $I_{\text{real}}^{(k)}$ over different orders $k$. Each row shows the change in the distribution during the training process. Experiments showed that the two-phase phenomenon widely existed on different DNNs trained on various datasets. It also verified the finding in [41] that the beginning of the 2nd phase was temporally aligned with the time point when the loss gap increased. Please see Appendix J.1 for results on the other six DNNs trained for 3D point cloud/image/sentiment classification.

• *Discrimination property.* This property means that the same interaction extracted from different samples consistently contributes to the classification of a certain category. This property has been observed on various DNNs [21], and it implies that interactions are discriminative for classification.

**Complexity/order of interactions.** The *complexity* (or *order*) of an interaction is defined as the number of input variables in the set $S$, *i.e.*, $\text{order}(S) \stackrel{\text{def}}{=} |S|$. In this way, a high-order interaction represents a complex non-linear relationship among many input variables.

### 3.2 Two-phase dynamics of learning interactions

Zhang et al. [41] have discovered the following two-phase dynamics of interaction complexity during the training process. (1) As Figure 2 shows, before the training process, the DNN with randomly initialized parameters mainly encodes interactions of medium orders. (2) In the first phase, the DNN removes initial interactions of medium and high orders, and mainly encodes low-order interactions. (3) In the second phase, the DNN gradually learns interactions of increasing orders.

To better illustrate this phenomenon, we followed [41] to conduct experiments on different DNNs, including AlexNet [17], VGG [31], BERT [9], DGCNN [38], and on various datasets, including image data (MNIST [19], CIFAR-10 [16], CUB-200-2011 [36], and Tiny-ImageNet [18]), natural language data (SST-2 [32]), and point cloud data (ShapeNet [39]). For image data, we followed [41] to select a random set of ten image patches as input variables. For natural language data, we set the entire embedding vector of each token as an input variable. For point cloud data, we took point clusters as input variables. Please see Appendix G.3 for the detailed settings. We set $v(\boldsymbol{x}) = \log\left(p(y^{\text{truth}}|\boldsymbol{x})/[1 - p(y^{\text{truth}}|\boldsymbol{x})]\right)$ by following [7], where $p(y^{\text{truth}}|\boldsymbol{x})$ denotes the probability of classifying the input sample $\boldsymbol{x}$ to the ground-truth category. We followed [41] to define the interaction whose absolute value is greater than or equal to $\tau = 0.03 \, \mathbb{E}_{\boldsymbol{x}}[|v(\boldsymbol{x}) - v(\boldsymbol{x}_\emptyset)|]$ as salient interaction. For interactions of each $k$-th order, we normalized the strength of salient interactions as $I_{\text{real}}^{(k)} = \mathbb{E}_{\boldsymbol{x}}[\sum_{\text{type} \in \{\text{and,or}\}} \sum_{S:|S|=k, |I_{\text{type}}(S|\boldsymbol{x})| \geq \tau} |I_{\text{type}}(S|\boldsymbol{x})|]/Z$ to enable fair comparison between different training epochs[9], where $Z = \mathbb{E}_{1 \leq k' \leq n} \mathbb{E}_{\boldsymbol{x}}[\sum_{\text{type} \in \{\text{and,or}\}} \sum_{S:|S|=k', |I_{\text{type}}(S|\boldsymbol{x})| \geq \tau} |I_{\text{type}}(S|\boldsymbol{x})|]$ denotes the normalizing constant.

Figure 2 shows how the distribution of interaction strength $I_{\text{real}}^{(k)}$ of different orders changed throughout the entire training process, and it demonstrates that the two-phase dynamics widely existed on different DNNs trained on various datasets. Before training, the interaction strength of medium orders dominated, and the distribution of interaction strength of different orders looked like a spindle. In the first phase (from the 2nd column to the 3rd column in the figure), the strength of medium-order and high-order interactions gradually shrank to zero, while the strength of low-order interactions

---

[9]The normalization removes the effect of the explosion of output values during the training process and enables us to only analyze the relative distribution of interaction strength.

increased. In the second phase (from the 3rd column to the 6th column in the figure), the DNN learned interactions of increasing orders (complexities).

**How to understand the two-phase phenomenon.** Previous studies [44, 26] have observed and partially proved that the complexity/order of an interaction can reflect the generalization ability[10] of the interaction. Let us consider an interaction that is frequently extracted by a DNN from training samples (see the transferability property in Section 3.2). If this interaction also frequently appears in testing samples, then this interaction is considered generalizable[10]; otherwise, non-generalizable. To this end, Zhou et al. [44] have discovered that high-order (complex) interactions are less generalizable between training and testing samples than low-order (simple) interactions. Furthermore, Ren et al. [26] have proved that high-order (complex) interactions are more unstable than low-order (simple) interactions when input variables or network parameters are perturbed by random noises.

Therefore, the two-phase dynamics enable us to revisit the change of generalization power of a DNN:

1. Before training, the interactions extracted from an initialized DNN exhibited a spindle-shaped distribution of interaction strength over different orders. These interactions could be considered random patterns irrelevant to the task, and such patterns were mostly of medium orders.

2. In the first phase, the DNN mainly removed the irrelevant patterns caused by the randomly initialized parameters. At the same time, the DNN shifted its attention to low-order interactions between very few input variables. These low-order interactions usually represented relatively simple and generalizable[10] inference patterns, without encoding complex inference patterns.

3. In the second phase, the DNN gradually learned interactions of increasing orders (increasing complexities). **Although there was no clear boundary between under-fitting and over-fitting in mathematics, the learning of very complex interactions had been widely considered as a typical sign of over-fitting**[10] [44].

### 3.3 Proving of the two-phase dynamics

#### 3.3.1 Analytic solution to interaction effects

As the foundation of proving the dynamics of the two phases, let us first derive the analytic solution to interaction effects at a specific time point during the training process. Then, Sections 3.3.2 and 3.3.3 will use this analytic solution to further explain detailed dynamics in the second phase and the first phase, respectively. Later experiments show that our theory can well predict the true dynamics of all AND-OR interactions during the learning of real DNNs.

The proof in this subsection can be divided into three steps. (1) We first rewrite a DNN's inference on an input sample as a weighted sum of triggering functions of different interactions. (2) Then, we can reformulate the learning of the DNN on an input sample as a linear regression problem. (3) Thus, the interactions at an intermediate time point during training can be obtained as the optimal solution to the linear regression problem under a certain level of parameter noises.

• **Step 1: Rewriting a DNN's inference on an input sample as a weighted sum of triggering functions of different interactions.** For simplicity, let us only focus on the dynamics of AND interactions, because OR interactions can also be represented as a specific kind of AND interactions[8] (see Appendix E for details). In this way, without loss of generality, let us just analyze the learning of AND interactions *w.r.t.* $v_{\text{and}}(\boldsymbol{x}) = v(\boldsymbol{x}_\emptyset) + \sum_{\emptyset \neq S \subseteq N} I_{\text{and}}(S|\boldsymbol{x})$, and simplify the notation as $v(\boldsymbol{x}) = v(\boldsymbol{x}_\emptyset) + \sum_{\emptyset \neq S \subseteq N} I(S|\boldsymbol{x})$ in the following proof. Our conclusions can also be extended to OR interactions, as mentioned above.

Given a DNN, we follow [26, 22] to rewrite the inference function of the network $v(\boldsymbol{x})$. This is inspired by the universal matching property of interactions in Theorem 2, *i.e.*, given any arbitrarily masked input sample $\hat{\boldsymbol{x}}_S$ *w.r.t.* a random subset $S \subseteq N$, the network output can always be represented as a linear sum of different interaction effects $v(\boldsymbol{x} = \hat{\boldsymbol{x}}_S) = \sum_{T \subseteq S} I(T|\boldsymbol{x} = \hat{\boldsymbol{x}})$. In this way, the following equation rewrites the inference function of the DNN $v(\boldsymbol{x} = \hat{\boldsymbol{x}}_S)$ as the weighted sum of triggering functions of interactions (see Appendix F.2 for proof).

$$\forall\, S \subseteq N,\; v(\boldsymbol{x} = \hat{\boldsymbol{x}}_S) = f(\boldsymbol{x} = \hat{\boldsymbol{x}}_S),\; \text{ subject to } f(\boldsymbol{x}) \stackrel{\text{def}}{=} \sum_{T \subseteq N} w_T\, J_T(\boldsymbol{x}), \quad (6)$$

---

[10]Unlike the traditional definition of the over-fitting/generalization power on the entire model over the entire dataset, the interaction first enables us to explicitly identify detailed over-fitted/generalizable inference patterns (interactions) on a specific sample.

where the interaction triggering function $J_T(\boldsymbol{x})$ is a real-valued approximation of the binary indicator function $\mathbb{1}(\hat{\boldsymbol{x}}_S$ triggers the AND relation $T)$ in Eq. (3) and returns the triggering value of the interaction pattern $T$. In particular, we set $w_\emptyset = v(\boldsymbol{x} = \hat{\boldsymbol{x}}_\emptyset)$, $J_\emptyset(\boldsymbol{x}) = 1$. $J_T(\boldsymbol{x})$ is computed as a sum of compositional terms in the Taylor expansion of $v(\boldsymbol{x})$.

$$J_T(\boldsymbol{x}) = \sum_{\boldsymbol{\pi} \in Q_T} \frac{1}{\prod_{i=1}^n \pi_i!} \frac{\partial^{\pi_1 + \cdots + \pi_n} v}{\partial x_1^{\pi_1} \cdots \partial x_n^{\pi_n}}\Big|_{\boldsymbol{x} = \boldsymbol{x}_\emptyset} \prod_{i \in T} (x_i - b_i)^{\pi_i} / w_T, \tag{7}$$

where the scalar weight $w_T$ should be computed as $w_T = I(T | \boldsymbol{x} = \hat{\boldsymbol{x}})$ to satisfy the equality in Eq. (6), and $Q_T = \{[\pi_1, \ldots, \pi_n]^\top : \forall i \in T, \pi_i \in \mathbb{N}^+; \forall i \notin T, \pi_i = 0\}$. See Appendix F.2 for proof.

**Understanding $J_T(\boldsymbol{x})$ and $w_T$.** Let us consider a masked sample $\hat{\boldsymbol{x}}_S$ in which input variables in $N \setminus S$ are masked. If $T \subseteq S$, which means all input variables in $T$ are not masked in $\hat{\boldsymbol{x}}_S$, then $J_T(\hat{\boldsymbol{x}}_S) = 1$, indicating the interaction pattern is triggered; otherwise, $J_T(\hat{\boldsymbol{x}}_S) = 0$, indicating the interaction pattern is not triggered. $w_T$ is a scalar weight. Particularly, let $I_f(T | \boldsymbol{x})$ denote the interaction extracted from the function $f(\boldsymbol{x}) = \sum_{T \subseteq N} w_T J_T(\boldsymbol{x})$, then we have $I_f(T | \boldsymbol{x}) = w_T$.

● **Step 2: Based on Eq. (6), the learning of the DNN on an input sample can be reformulated as learning the scalar weight $w_T$ for each interaction triggering function $J_T(\boldsymbol{x})$, under a linear regression setting.** We can roughly consider the learning problem as a linear regression to a set of *potentially true interactions*, because it has been discovered by [21, 4] that different DNNs for the same task usually encode similar sets of interactions. Therefore, the learning of a DNN can be considered as training a model to fit a set of pre-defined interactions. *In spite of the above simplifying settings, subsequent experiments in Figure 4 still verify that our theoretical results can well predict the learning dynamics of interactions in real DNNs.*

Specifically, let the DNN be trained on a set of samples $\mathcal{D} = \{(\boldsymbol{x}, y)\}$. According to Theorem 2, given each training sample $\boldsymbol{x}$, output scores of the finally converged DNN on all $2^n$ randomly masked samples $\{\boldsymbol{x}_S : S \subseteq N\}$ can be written in the form of $y_S \overset{\text{def}}{=} y(\boldsymbol{x}_S) = v(\boldsymbol{x}_\emptyset) + \sum_{\emptyset \neq T \subseteq N} \mathbb{1}(\boldsymbol{x}_S \text{ triggers interaction } T) \cdot w_T^* = v(\boldsymbol{x}_\emptyset) + \sum_{\emptyset \neq T \subseteq S} w_T^*$, which is determined by parameters $\{w_T^* : T \subseteq N\}$[11]. $\{w_T^* : T \subseteq N\}$ can be taken as a set of true interactions that the DNN needs to learn. Therefore, the learning of the converged interactions on the training sample $\boldsymbol{x}$ can be represented as the regression towards the converged function $y(\boldsymbol{x}_S)$ on all masked samples $\{(\boldsymbol{x}_S, y_S) : S \subseteq N\}$.

$$L(\boldsymbol{w}) = \mathbb{E}_{S \subseteq N}[(y_S - \boldsymbol{w}^\top \boldsymbol{J}(\boldsymbol{x}_S))^2]. \tag{8}$$

where we simplify the notation as follows. $\boldsymbol{w} \overset{\text{def}}{=} \text{vec}(\{w_T : T \subseteq N\}) \in \mathbb{R}^{2^n}$ denotes the weight vector of $2^n$ different interactions, and $\boldsymbol{J}(\boldsymbol{x}_S) \overset{\text{def}}{=} \text{vec}(\{J_T(\boldsymbol{x}_S) : T \subseteq N\}) \in \mathbb{R}^{2^n}$ denotes the vector of triggering values of $2^n$ different interactions $\{T \subseteq N\}$ on the masked sample $\boldsymbol{x}_S$.

● **Step 3: Directly optimizing Eq. (8) gives the interactions of the finally converged DNN $w_T \leftarrow w_T^*$, but how do we estimate the interactions in an intermediate time point during the training process?** To this end, we assume that the training process of the DNN is subject to parameter noises (see Lemma 1). In fact, this assumption is common. Before training, randomly initialized parameters in the DNN are pure noises without clear meanings. In this way, the DNN's training process can be viewed as a process of gradually reducing the noise on its parameters. This is also supported by the lottery ticket hypothesis [10], *i.e.*, the learning process actually penalizes most noisy parameters and learns a very small number of meaningful parameters. *Therefore, as training proceeds, the noise on the network parameters can be considered to gradually diminish.*

**Lemma 1** (Noisy triggering function, proven in Appendix F.3). *If the inference score of the DNN contains an unlearnable noise, i.e., $\forall S \subseteq N, \widetilde{v}(\boldsymbol{x}_S) = v(\boldsymbol{x}_S) + \Delta v_S, \Delta v_S \sim \mathcal{N}(0, \sigma^2)$, then the interaction between input variables w.r.t. $\emptyset \neq T \subseteq N$, extracted from inference scores $\{\widetilde{v}(\boldsymbol{x}_S)\}$ can be written as $\widetilde{I}(T | \boldsymbol{x}) = I(T | \boldsymbol{x}) + \Delta I_T$, where $\Delta I_T$ denotes the noise in the interaction caused by the noise in the output $\Delta v_S$, and we have $\mathbb{E}[\Delta I_T] = 0$ and $\text{Var}[\Delta I_T] = 2^{|T|} \sigma^2$. In this way, given an input sample $\hat{\boldsymbol{x}}$, we can consider the scalar weight $w_T = I(T | \boldsymbol{x} = \hat{\boldsymbol{x}})$, and consider the interaction triggering function $\widetilde{J}_T(\boldsymbol{x}) = J_T(\boldsymbol{x}) + \epsilon_T$, where $J_T(\boldsymbol{x})$ is defined in Eq. (7). $\epsilon_T = \Delta I_T / w_T$ represents the noise term on the triggering function. We have $\mathbb{E}[\epsilon_T] = 0$ and $\text{Var}[\epsilon_T] \propto 2^{|T|} \sigma^2$ w.r.t. noises.*

---

[11]Note that in the converged output $y_S$, the true interactions $\{w_T^* : T \subseteq N\}$ actually mean interactions extracted from the finally converged DNN, which probably contain over-fitted interaction patterns. *I.e.*, $\{w_T^* : T \subseteq N\}$ is not the ideal representation for the task.

Therefore, the learned interactions under unavoidable parameter noises can be represented as minimizing the following loss, where we vectorize the noise $\boldsymbol{\epsilon} = \text{vec}(\{\epsilon_T : T \subseteq N\}) \in \mathbb{R}^{2^n}$ for simplicity.

$$\widetilde{L}(\boldsymbol{w}) = \mathbb{E}_{\boldsymbol{\epsilon}}\mathbb{E}_{S \subseteq N}[(y_S - \boldsymbol{w}^\top \widetilde{\boldsymbol{J}}(\boldsymbol{x}_S))^2] = \mathbb{E}_{\boldsymbol{\epsilon}}\mathbb{E}_{S \subseteq N}[(y_S - \boldsymbol{w}^\top (\boldsymbol{J}(\boldsymbol{x}_S) + \boldsymbol{\epsilon}))^2]. \qquad (9)$$

**Remark.** The minimizer to Eq. (9) **does not** represent the end of training, but represents the *intermediate state* of interactions after *a certain epoch in the training process*. We formulate the training process as a process of gradually reducing the noise on the DNN's parameters, and the minimizer $\hat{\boldsymbol{w}}$ to Eq. (9) represents the optimal interaction state when the training is subject to certain *parameter noises*. We will show later that the minimizer $\hat{\boldsymbol{w}}$ computed under different noise levels can accurately predict the dynamics of interactions during the training process (see Figures 4 and 8).

**Assumption 1.** *To simplify the proof, we assume that different noise terms $\epsilon_T$ on the triggering function are independent, and uniformly set the variance as $\forall\, T \subseteq N$, $\text{Var}[\epsilon_T] = 2^{|T|}\sigma^2$.*

Assumption 1 is made according to two findings in Lemma 1: (1) the interaction triggering function $\widetilde{J}_T(\boldsymbol{x})$ is real-valued subject to the noise on the DNN's parameters, (2) the variance of the interaction triggering function $\widetilde{J}_T(\boldsymbol{x})$ increases exponentially along with the order $|T|$. More importantly, the assumed exponential increase of the variance in the above finding (2) has been widely observed in various DNNs trained for different tasks in previous experiments [26, 22].

**Theorem 3** (Proven in Appendix F.4). *Let $\hat{\boldsymbol{w}} = \arg\min_{\boldsymbol{w}} \widetilde{L}(\boldsymbol{w})$ denote the optimal solution to the minimization of the loss function $\widetilde{L}(\boldsymbol{w})$. Then, we have*

$$\hat{\boldsymbol{w}} = (\boldsymbol{J}^\top \boldsymbol{J} + 2^n \text{diag}(\boldsymbol{c}))^{-1}\boldsymbol{J}^\top \boldsymbol{y} = (\boldsymbol{J}^\top \boldsymbol{J} + 2^n \text{diag}(\boldsymbol{c}))^{-1}\boldsymbol{J}^\top \boldsymbol{J}\boldsymbol{w}^* = \hat{\boldsymbol{M}}\boldsymbol{w}^*, \qquad (10)$$

*where $\boldsymbol{J} \stackrel{\text{def}}{=} [\boldsymbol{J}(\boldsymbol{x}_{S_1}), \boldsymbol{J}(\boldsymbol{x}_{S_2}), \cdots, \boldsymbol{J}(\boldsymbol{x}_{S_{2^n}})]^\top \in \mathbb{R}^{2^n \times 2^n}$ is a matrix to represent the triggering values of $2^n$ interactions (w.r.t. $2^n$ columns) on $2^n$ masked samples (w.r.t. $2^n$ rows). $\boldsymbol{x}_{S_1}, \boldsymbol{x}_{S_2}, \cdots, \boldsymbol{x}_{S_{2^n}}$ enumerate all masked samples. $\boldsymbol{y} \stackrel{\text{def}}{=} [y(\boldsymbol{x}_{S_1}), y(\boldsymbol{x}_{S_2}), \cdots, y(\boldsymbol{x}_{S_{2^n}})]^\top \in \mathbb{R}^{2^n}$ enumerates the finally-converged outputs on $2^n$ masked samples. $\boldsymbol{c} \stackrel{\text{def}}{=} \text{vec}(\{\text{Var}[\epsilon_T] : T \subseteq N\}) = \text{vec}(\{2^{|T|}\sigma^2 : T \subseteq N\}) \in \mathbb{R}^{2^n}$ denotes the vector of variances of the triggering values of $2^n$ interactions. The matrix $\hat{\boldsymbol{M}}$ is defined as $\hat{\boldsymbol{M}} \stackrel{\text{def}}{=} (\boldsymbol{J}^\top \boldsymbol{J} + 2^n \text{diag}(\boldsymbol{c}))^{-1}\boldsymbol{J}^\top \boldsymbol{J}$, and $\boldsymbol{w}^* \stackrel{\text{def}}{=} \text{vec}(\{w_T^* : T \subseteq N\})$.*

In this way, Theorem 3 provides an analytic solution to the minimization of $\widetilde{L}(\boldsymbol{w})$ under parameter noises. Experiments in Figure 4 will show that the learning dynamics of interactions derived from our simplifying assumption can still predict the real distribution of interactions over different orders.

### 3.3.2 Explaining the dynamics in the second phase

Based on the above analytic solution, this subsection aims to prove that in the second phase, the DNN first encodes interactions of low orders and then gradually encodes interactions of higher orders.

• **The second phase can be viewed as a process of gradually reducing the noise level $\sigma^2$.** The analytic solution $\hat{\boldsymbol{w}}$ in Theorem 3 under different noise levels $\sigma^2$ enables us to analyze the dynamics of interactions during the second phase. This is because the noise on the network parameters can be considered to gradually diminish during the training process, as we assume in Section 3.3.1. Then accordingly, the noise level $\sigma^2$ of the noise term $\epsilon_T$ on the interaction triggering function also gradually diminishes during training. At the start of the second phase, the noise level $\sigma^2$ is large, and the interaction triggering function $\widetilde{J}_T(\boldsymbol{x})$ is dominated by the noise term $\epsilon_T$. Later, as training proceeds in the second phase, the noise level $\sigma^2$ gradually decreases, making less effect on the interaction triggering function.

• **The change of the analytic solution $\hat{\boldsymbol{w}}$ along with the decreasing noises $\sigma^2$ explains the dynamics in the second phase.** We prove that as $\sigma^2$ decreases, the ratio of low-order interaction strength to high-order interaction strength in the analytic solution $\hat{\boldsymbol{w}}$ decreases. This means that the DNN gradually learns higher-order interactions in the second phase, which can be verified by our observation in Figure 2. The detailed results are derived as follows.

**Lemma 2** (Proven in Appendix F.5). *The compositional term $J_T(\boldsymbol{x})$ in the Taylor expansion in Eq. (7) always has fixed values on $2^n$ masked samples $\{\boldsymbol{x}_S : S \subseteq N\}$, i.e., $\forall S \subseteq N$, $J_T(\boldsymbol{x}_S) = \mathbb{1}(T \subseteq S)$. It means that the matrix $\boldsymbol{J} = [\boldsymbol{J}(\boldsymbol{x}_{S_1}), \boldsymbol{J}(\boldsymbol{x}_{S_2}), \cdots, \boldsymbol{J}(\boldsymbol{x}_{S_{2^n}})]^\top \in \{0, 1\}^{2^n \times 2^n}$ in Eq. (10) is a fixed binary matrix, no matter how we change the DNN $v(\cdot)$ or the input sample $\boldsymbol{x}$.*

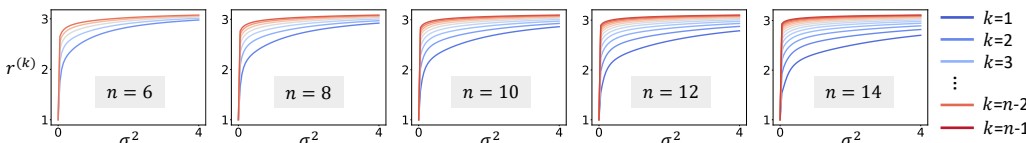

Figure 3: Monotonic increase of $r^{(k)}$ along with $\sigma^2$ mentioned in Proposition 1. We show the curves of $r^{(k)}$ when we set different numbers of input variables $n$ and different orders $k = 1, \cdots, n-1$.

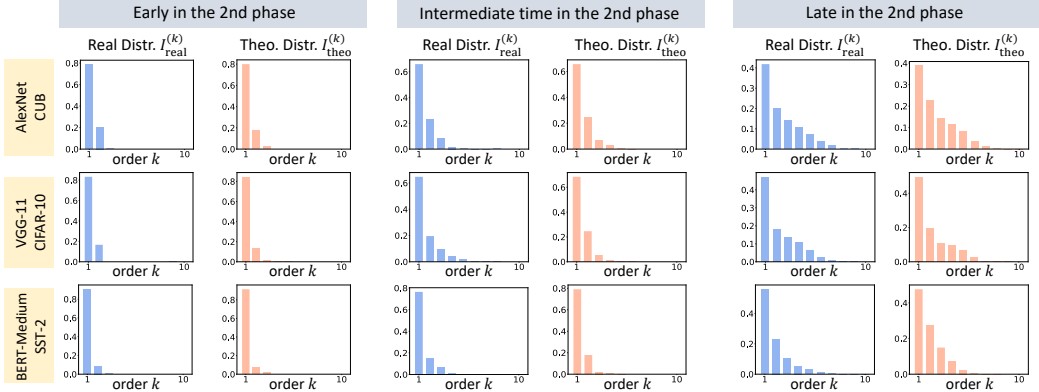

Figure 4: Comparison between the theoretical distribution of interaction strength $I_{\text{theo}}^{(k)}$ and the real distribution of interaction strength $I_{\text{real}}^{(k)}$ in the second phase. Please see Appendix J.3 for the comparison on the other six DNNs trained for 3D point cloud/image/sentiment classification.

**Theorem 4** (Proven in Appendix F.6). *According to Theorem 3, we can write the analytic solution of the interaction effect $\hat{w}_T$ w.r.t. a subset $T$ as $\hat{w}_T = \hat{\boldsymbol{m}}_T^\top \boldsymbol{w}^*$, where $\hat{\boldsymbol{m}}_T^\top \in \mathbb{R}^{1 \times 2^n}$ denotes a row vector of the matrix $\hat{\boldsymbol{M}} = [\hat{\boldsymbol{m}}_{T_1}, \hat{\boldsymbol{m}}_{T_2} \cdots, \hat{\boldsymbol{m}}_{T_{2^n}}]^\top$, indexed by $T$. Combining with Lemma 2, for any two subsets $T, T' \subseteq N$ of the same order, i.e., $|T| = |T'|$, we have $\|\hat{\boldsymbol{m}}_T\|_2 = \|\hat{\boldsymbol{m}}_{T'}\|_2$.*

**Proposition 1.** *For any two subsets $T, T' \subseteq N$ with $|T| < |T'|$, $\|\hat{\boldsymbol{m}}_T\|_2/\|\hat{\boldsymbol{m}}_{T'}\|_2$ is greater than 1 and decreases monotonically as $\sigma^2$ decreases throughout training. The norm $\|\hat{\boldsymbol{m}}_T\|_2$ is only determined by $n$, $\sigma^2$, and the order $|T|$, but is agnostic to finally-converged interactions $\{w_T^* : T \subseteq N\}$.*

Proposition 1 shows a monotonic decrease of $\|\hat{\boldsymbol{m}}_T\|_2/\|\hat{\boldsymbol{m}}_{T'}\|_2$ along with the decrease of $\sigma^2$. The physical meaning of $\|\hat{\boldsymbol{m}}_T\|_2/\|\hat{\boldsymbol{m}}_{T'}\|_2$ can be understood as follows. According to $\hat{w}_T = \hat{\boldsymbol{m}}_T^\top \boldsymbol{w}^*$, $\|\hat{\boldsymbol{m}}_T\|_2$ reflects the strength of the DNN encoding the interaction $T$. In this way, $\|\hat{\boldsymbol{m}}_T\|_2/\|\hat{\boldsymbol{m}}_{T'}\|_2$ measures the relative strength of encoding a low-order interaction $T$ w.r.t. that of encoding a high-order interaction $T'$.

***Conclusions from Theorem 4 and Proposition 1:* Because the second phase is viewed as a process of gradually reducing the noise level $\sigma^2$, Theorem 4 and Proposition 1 explain why the DNN mainly encodes low-order interactions and suppresses high-order interactions at the start of the second phase (when $\sigma^2$ is large). They also explain why the DNN learns interactions of increasing orders during the second phase (when $\sigma^2$ gradually decreases).**

*Experimental verification of Proposition 1:* We measured the relative strength $r^{(k)} \stackrel{\text{def}}{=} \|\hat{\boldsymbol{m}}_T\|_2/\|\hat{\boldsymbol{m}}_{T'}\|_2$ subject to $|T| = k$ and $|T'| = k+1$, for $k = 1, \cdots, n-1$, under different values of $\sigma^2$. Figure 3 shows that when $\sigma^2$ decreased, $r^{(k)}$ monotonically decreased for all orders $k = 1, \cdots, n-1$, which verified the proposition. The experiment was conducted using different numbers of input variables $n$.

**Theorem 5** (Proven in Appendix F.7). *When $\sigma = 0$, $\hat{\boldsymbol{w}}$ satisfies $\forall \emptyset \neq T \subseteq N$, $\hat{w}_T = w_T^*$.*

Theorem 5 shows a special case when there is no noise on the network parameters. Then, the DNN learns the finally converged interactions $\{w_T^* : T \subseteq N\}$. Note that the finally converged DNN probably encodes some interactions of high orders, which correspond to over-fitted patterns.

• **Experiments on real datasets.** We conducted experiments to examine whether our theory could predict the real dynamics of interaction strength of different orders when we trained DNNs in practice. We trained AlexNet and VGG on the MNIST dataset, the CIFAR-10 dataset, the CUB-200-2011

dataset, and the Tiny-ImageNet dataset, trained BERT-Tiny and BERT-Medium on the SST-2 dataset, and trained DGCNN on the ShapeNet dataset. Then, we computed the real distribution of interaction strength over different orders on each DNN, and tracked the change of the distribution throughout the training process. As mentioned in Section 3.2, the real interaction strength of each $k$-th order was quantified as $I_{\text{real}}^{(k)} = \mathbb{E}_{\boldsymbol{x}}[\sum_{S:|S|=k,|I(S|\boldsymbol{x})|\geq\tau}|I(S|\boldsymbol{x})|] / Z$ [12]. Accordingly, we defined the metric $I_{\text{theo}}^{(k)} = \mathbb{E}_{\boldsymbol{x}}[\sum_{S:|S|=k,|\hat{w}_S|\geq\tau_{\text{theo}}}|\hat{w}_S|] / Z_{\text{theo}}$ in the same way of $I_{\text{real}}^{(k)}$ to measure the theoretical distribution of the interaction strength, where $Z_{\text{theo}} = \mathbb{E}_{1\leq k'\leq n}\mathbb{E}_{\boldsymbol{x}}[\sum_{S:|S|=k',|\hat{w}_S|\geq\tau_{\text{theo}}}|\hat{w}_S|]$, $\tau_{\text{theo}} = 0.03 \cdot |v_{\text{theo}}(\boldsymbol{x}) - \hat{w}_{\emptyset}|$, and $v_{\text{theo}}(\boldsymbol{x}) \overset{\text{def}}{=} \sum_{S\subseteq N}\hat{w}_S$. To compute the theoretical solution $\hat{\boldsymbol{w}} = \hat{\boldsymbol{M}}\boldsymbol{w}^*$ in Eq. (10), given an input sample $\boldsymbol{x}$, we used the set of salient interactions $\Omega = \{S \subseteq N : |I(S|\boldsymbol{x})| \geq \tau\}$) extracted from the finally converged DNN to construct the set of true interactions $\boldsymbol{w}^*$.

Figure 4 shows that the theoretical distribution $I_{\text{theo}}^{(k)}$ could well match the real distribution $I_{\text{real}}^{(k)}$ at different training epochs. Particularly, we used a sequence of theoretical distributions of $I_{\text{theo}}^{(k)}$ with decreasing $\sigma^2$ values to match the real distribution of $I_{\text{real}}^{(k)}$ at different epochs. The $\sigma^2$ value was determined to achieve the best match between $I_{\text{theo}}^{(k)}$ and $I_{\text{real}}^{(k)}$.

### 3.3.3 Explaining the dynamics in the first phase

Because the spindle-shaped distribution of interaction strength in a randomly initialized DNN has already been proven by [41], in this subsection, let us further explain the DNN's dynamics in the first phase based on Eq. (9). As previously shown in Figure 2, in the first phase, the DNN removes initial interactions of medium and high orders, and mainly encodes low-order interactions.

Therefore, the first phase is explained as the process of removing chaotic initial interactions and converging to the optimal solution to Eq. (9) under large parameter noise (*i.e.*, large $\sigma^2$). In sum, the first phase is a process of *pushing initial random interactions to the optimal solution*, while the second phase corresponds to the *change of the optimal solution* as $\sigma^2$ gradually decreases.

## 4 Conclusion and discussion

In this study, we have proven the two-phase dynamics of a DNN learning interactions of different orders. Specifically, we have followed [26, 22] to reformulate the learning of interactions as a linear regression problem on a set of interaction triggering functions. In this way, we have successfully derived an analytic solution to interaction effects when the DNN was learned with unavoidable parameter noises. This analytic solution has successfully predicted a DNN's two-phase dynamics of learning interactions in real experiments. Considering a series of recent theoretical guarantees of taking interactions as faithful primitive inference patterns encoded by the DNN [44, 27], our study has first mathematically explained why and how the learning process gradually shifts attention from generalizable (low-order) inference patterns to probably over-fitted (high-order) inference patterns.

**Practical implications.** A theoretical understanding of the two-phase dynamics of interactions offers a new perspective to monitor the overfitting level of the DNN on different training samples throughout training. The two-phase dynamics enables us to evaluate the overfitting level of each specific sample, making overfitting no longer a problem *w.r.t.* the entire dataset. We can track the change of the interaction complexity for each training sample, and take the time point when high-order interactions increase as a sign of overfitting. In this way, the two-phase dynamics of interactions may help people remove overfitted samples from training and guide the early stopping on a few "hard samples."

**Acknowledgements.** This work is partially supported by the National Science and Technology Major Project (2021ZD0111602), the National Nature Science Foundation of China (92370115, 62276165). This work is also partially supported by Huawei Technologies Inc.

---

[12]In experiments, the real distribution of interaction strength $I_{\text{real}}^{(k)}$ was computed using both AND and OR interactions. Because the OR interaction was a special AND interaction and had similar dynamics, this experiment actually tested the fidelity of our theory to explain the dynamics of all interactions. Nevertheless, Appendix J.4 also reports the fitness of the theoretical distribution $I_{\text{theo}}^{(k)}$ and real distribution of AND interactions.

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

# A  Properties of the AND interaction

The Harsanyi interaction [14] (*i.e.*, the AND interaction in this paper) was a standard metric to measure the AND relationship between input variables encoded by the network. In this section, we present several desirable properties/axioms that the Harsanyi AND interaction $I_{\text{and}}(S|\boldsymbol{x})$ satisfies. These properties further demonstrate the faithfulness of using Harsanyi AND interaction to explain the inference score of a DNN.

(1) *Efficiency axiom* (proven by [14]). The output score of a model can be decomposed into interaction effects of different patterns, *i.e.* $v(\boldsymbol{x}) = \sum_{S \subseteq N} I_{\text{and}}(S|\boldsymbol{x})$.

(2) *Linearity axiom*. If we merge output scores of two models $v_1$ and $v_2$ as the output of model $v$, *i.e.* $\forall S \subseteq N,\ v(\boldsymbol{x}_S) = v_1(\boldsymbol{x}_S) + v_2(\boldsymbol{x}_S)$, then their interaction effects $I_{\text{and}}^{v_1}(S|\boldsymbol{x})$ and $I_{\text{and}}^{v_2}(S|\boldsymbol{x})$ can also be merged as $\forall S \subseteq N, I_{\text{and}}^{v}(S|\boldsymbol{x}) = I_{\text{and}}^{v_1}(S|\boldsymbol{x}) + I_{\text{and}}^{v_2}(S|\boldsymbol{x})$.

(3) *Dummy axiom*. If a variable $i \in N$ is a dummy variable, *i.e.* $\forall S \subseteq N \setminus \{i\}, v(\boldsymbol{x}_{S \cup \{i\}}) = v(\boldsymbol{x}_S) + v(\boldsymbol{x}_{\{i\}})$, then it has no interaction with other variables, $\forall \emptyset \neq S \subseteq N \setminus \{i\}, I_{\text{and}}(S \cup \{i\}|\boldsymbol{x}) = 0$.

(4) *Symmetry axiom*. If input variables $i, j \in N$ cooperate with other variables in the same way, $\forall S \subseteq N \setminus \{i, j\}, v(\boldsymbol{x}_{S \cup \{i\}}) = v(\boldsymbol{x}_{S \cup \{j\}})$, then they have same interaction effects with other variables, $\forall S \subseteq N \setminus \{i, j\}, I_{\text{and}}(S \cup \{i\}|\boldsymbol{x}) = I_{\text{and}}(S \cup \{j\}|\boldsymbol{x})$.

(5) *Anonymity axiom*. For any permutations $\pi$ on $N$, we have $\forall S \subseteq N, I_{\text{and}}^{v}(S|\boldsymbol{x}) = I_{\text{and}}^{\pi v}(\pi S|\boldsymbol{x})$, where $\pi S \overset{\text{def}}{=} \{\pi(i)|i \in S\}$, and the new model $\pi v$ is defined by $(\pi v)(\boldsymbol{x}_{\pi S}) = v(\boldsymbol{x}_S)$. This indicates that interaction effects are not changed by permutation.

(6) *Recursive axiom*. The interaction effects can be computed recursively. For $i \in N$ and $S \subseteq N \setminus \{i\}$, the interaction effect of the pattern $S \cup \{i\}$ is equal to the interaction effect of $S$ with the presence of $i$ minus the interaction effect of $S$ with the absence of $i$, *i.e.* $\forall S \subseteq N \setminus \{i\}, I_{\text{and}}(S \cup \{i\}|\boldsymbol{x}) = I_{\text{and}}(S|\boldsymbol{x}, i \text{ is always present}) - I_{\text{and}}(S|\boldsymbol{x})$. $I_{\text{and}}(S|\boldsymbol{x}, i \text{ is always present})$ denotes the interaction effect when the variable $i$ is always present as a constant context, *i.e.* $I_{\text{and}}(S|\boldsymbol{x}, i \text{ is always present}) = \sum_{L \subseteq S}(-1)^{|S|-|L|} \cdot v(\boldsymbol{x}_{L \cup \{i\}})$.

(7) *Interaction distribution axiom*. This axiom characterizes how interactions are distributed for "interaction functions" [33]. An interaction function $v_T$ parameterized by a subset of variables $T$ is defined as follows. $\forall S \subseteq N$, if $T \subseteq S$, $v_T(\boldsymbol{x}_S) = c$ ; otherwise, $v_T(\boldsymbol{x}_S) = 0$. The function $v_T$ models pure interaction among the variables in $T$, because only if all variables in $T$ are present, the output value will be increased by $c$. The interactions encoded in the function $v_T$ satisfies $I_{\text{and}}(T|\boldsymbol{x}) = c$, and $\forall S \neq T, I_{\text{and}}(S|\boldsymbol{x}) = 0$.

# B  Common conditions for sparse interactions

Ren et al. [27] have formulated three mathematical conditions for the sparsity of AND interactions, as follows.

**Condition 1.** *The DNN does not encode interactions higher than the $M$-th order:* $\forall S \in \{S \subseteq N \mid |S| \geq M + 1\}, I_{and}(S|\boldsymbol{x}) = 0$.

Condition 1 implies that the DNN does not encode extremely high-order interactions. This is because extremely high-order interactions usually represent very complex and over-fitted patterns, which are unnecessary and unlikely to be learned by the DNN in real applications.

**Condition 2.** *Let us consider the average network output* $\bar{u}^{(k)} \overset{\text{def}}{=} \mathbb{E}_{|S|=k}[v(\boldsymbol{x}_S) - v(\boldsymbol{x}_\emptyset)]$ *over all masked samples* $\boldsymbol{x}_S$ *with $k$ unmasked input variables. This average network output monotonically increases when $k$ increases:* $\forall k' \leq k$, *we have* $\bar{u}^{(k')} \leq \bar{u}^{(k)}$.

Condition 2 implies that a well-trained DNN is likely to have higher classification confidence for input samples that are less masked.

**Condition 3.** *Given the average network output* $\bar{u}^{(k)}$ *of samples with $k$ unmasked input variables, there is a polynomial lower bound for the average network output of samples with $k'(k' \leq k)$ unmasked input variables:* $\forall k' \leq k$, $\bar{u}^{(k')} \geq (\frac{k'}{k})^p \bar{u}^{(k)}$, *where $p > 0$ is a positive constant.*

Condition 3 implies that the classification confidence of the DNN does not significantly degrade on masked input samples. The classification/detection of masked/occluded samples is common in real

scenarios. In this way, a well-trained DNN usually learns to classify a masked input sample based on local information (which can be extracted from unmasked parts of the input) and thus should not yield a significantly low confidence score on masked samples.

## C   Details of optimizing $\{\gamma_T\}$ to extract the sparsest AND-OR interactions

A method is proposed [20, 4] to simultaneously extract AND interactions $I_{\text{and}}(S|\boldsymbol{x})$ and OR interactions $I_{\text{or}}(S|\boldsymbol{x})$ from the network output. Given a masked sample $\boldsymbol{x}_T$, [20] proposed to learn a decomposition $v(\boldsymbol{x}_T) = v_{\text{and}}(\boldsymbol{x}_T) + v_{\text{or}}(\boldsymbol{x}_T)$ towards the sparsest interactions. The component $v_{\text{and}}(\boldsymbol{x}_T)$ was explained by AND interactions, and the component $v_{\text{or}}(\boldsymbol{x}_T)$ was explained by OR interactions. Specifically, they decomposed $v(\boldsymbol{x}_T)$ into $v_{\text{and}}(\boldsymbol{x}_T) = 0.5\, v(\boldsymbol{x}_T) + \gamma_T$ and $v_{\text{and}}(\boldsymbol{x}_T) = 0.5 \cdot v(\boldsymbol{x}_T) - \gamma_T$, where $\{\gamma_T : T \subseteq N\}$ is a set of learnable variables that determine the decomposition. In this way, the AND interactions and OR interactions can be computed according to Eq. (2), *i.e.*, $I_{\text{and}}(S|\boldsymbol{x}) = \sum_{T \subseteq S}(-1)^{|S|-|T|} v_{\text{and}}(\boldsymbol{x}_T)$, and $I_{\text{or}}(S|\boldsymbol{x}) = -\sum_{T \subseteq S}(-1)^{|S|-|T|} v_{\text{or}}(\boldsymbol{x}_{N \setminus T})$.

The parameters $\{\gamma_T\}$ were learned by minimizing the following LASSO-like loss to obtain sparse interactions:

$$\min_{\{\gamma_T\}} \sum_{S \subseteq N} |I_{\text{and}}(S|\boldsymbol{x})| + |I_{\text{or}}(S|\boldsymbol{x})| \tag{11}$$

**Removing small noises.** A small noise $\delta_S$ in the network output may significantly affect the extracted interactions, especially for high-order interactions. Thus, [20] proposed to learn to remove a small noise term $\delta_S$ from the computation of AND-OR interactions. Specifically, the decomposition was rewritten as $v_{\text{and}}(\boldsymbol{x}_T) = 0.5(v(\boldsymbol{x}_T) - \delta_T) + \gamma_T$ and $v_{\text{or}}(\boldsymbol{x}_T) = 0.5(v(\boldsymbol{x}_T) - \delta_T) + \gamma_T$. Thus, the parameters $\{\delta_T\}$, and $\{\gamma_T\}$ are simultaneously learned by minimizing the loss function in Eq. (11). The values of $\{\delta_T\}$ were constrained in $[-\zeta, \zeta]$ where $\zeta = 0.02 \cdot |v(\boldsymbol{x}) - v(\boldsymbol{x}_\emptyset)|$.

## D   Where does the coefficient $(-1)^{|S|-|T|}$ in Eq. (2) come from?

In fact, it is proven in [13] and [23] that the coefficient $(-1)^{|S|-|T|}$ in Eq. (2) is the **unique** coefficient to ensure that the interaction satisfies the **universal matching property**. Recall that the universal matching property means that no matter how we randomly mask an input sample $\boldsymbol{x}$, the network output on the masked sample $\boldsymbol{x}_S$ can always be accurately mimicked by the sum of interaction effects within $S$. An extension of this property for AND-OR interactions is also mentioned in Theorem 2.

## E   OR interactions can be considered a special kind of AND interactions

The OR interaction can be considered a specific kind of AND interaction, when we flip the masked state and presence (unmasked) state of each input variable.

Given an input sample $\boldsymbol{x} \in \mathbb{R}^n$, let $\boldsymbol{x}_T$ denote the masked sample obtained by masking input variables in $N \setminus T$, while leaving variables in $T$ unchanged. Specifically, the baseline values $\boldsymbol{b} \in \mathbb{R}^n$ are used to mask the input variables, which represent the masked states of the input variables. The definition of $\boldsymbol{x}_T$ is given as follows.

$$(\boldsymbol{x}_T)_i = \begin{cases} x_i, & i \in T \\ b_i, & i \in N \setminus T \end{cases} \tag{12}$$

Based on the above definition, the AND interaction is computed as $I_{\text{and}}(S|\boldsymbol{x}) = \sum_{T \subseteq S}(-1)^{|S|-|T|} v_{\text{and}}(\boldsymbol{x}_T)$, while the OR interaction is computed as $I_{\text{or}}(S|\boldsymbol{x}) = -\sum_{T \subseteq S}(-1)^{|S|-|T|} v_{\text{or}}(\boldsymbol{x}_{N \setminus T})$. To simplify the analysis, let us assume $v_{\text{and}}(\cdot) = v_{\text{or}}(\cdot) = 0.5 v(\cdot)$.

Then, let us consider a masked sample $\tilde{x}_T$, where we flip the masked state and presence (unmasked) state of each input variable. In this way, $\tilde{\boldsymbol{x}}_T$ is defined as follows.

$$(\tilde{\boldsymbol{x}}_T)_i = \begin{cases} x_i, & i \in N \setminus T \\ b_i, & i \in T \end{cases} \tag{13}$$

Therefore, the OR interaction $I_{\text{or}}(S|\boldsymbol{x})$ in Eq. 2 in main paper can be represented as an AND interaction $I_{\text{or}}(S|\tilde{\boldsymbol{x}})$, as follows.

$$I_{\text{or}}(S|\boldsymbol{x}) = -\sum_{T\subseteq S}(-1)^{|S|-|T|}v(\boldsymbol{x}_{N\setminus T}), \tag{14}$$

$$= -\sum_{T\subseteq S}(-1)^{|S|-|T|}v(\tilde{\boldsymbol{x}}_T), \tag{15}$$

$$= -I_{\text{and}}(S|\tilde{\boldsymbol{x}}). \tag{16}$$

In this way, the proof of the sparsity of AND interactions in [27] can also extend to OR interactions. Furthermore, we can simplify our analysis of the DNN's learning of interactions by only focusing on AND interactions.

## F    Proof of theorems

### F.1    Proof of Theorem 2

*Proof.* **(1) Universal matching theorem of AND interactions.**

We will prove that output component $v_{\text{and}}(\boldsymbol{x}_S)$ on all $2^n$ masked samples $\{\boldsymbol{x}_S : S \subseteq N\}$ could be universally explained by the all interactions in $S \subseteq N$, *i.e.*, $\forall \emptyset \neq S \subseteq N, v_{\text{and}}(\boldsymbol{x}_S) = \sum_{\emptyset \neq T \subseteq S}I_{\text{and}}(T|\boldsymbol{x}) + v(\boldsymbol{x}_\emptyset)$. In particular, we define $v_{\text{and}}(\boldsymbol{x}_\emptyset) = v(\boldsymbol{x}_\emptyset)$ (*i.e.*, we attribute output on an empty sample to AND interactions).

Specifically, the AND interaction is defined as $I_{\text{and}}(T|\boldsymbol{x}) = \sum_{L\subseteq T}(-1)^{|T|-|L|}v_{\text{and}}(\boldsymbol{x}_L)$ in 2. To compute the sum of AND interactions $\sum_{\emptyset \neq T \subseteq S}I_{\text{and}}(T|\boldsymbol{x}) = \sum_{\emptyset \neq T \subseteq S}\sum_{L\subseteq T}(-1)^{|T|-|L|}v_{\text{and}}(\boldsymbol{x}_L)$, we first exchange the order of summation of the set $L \subseteq T \subseteq S$ and the set $T \supseteq L$. That is, we compute all linear combinations of all sets $T$ containing $L$ with respect to the model outputs $v_{\text{and}}(\boldsymbol{x}_L)$ given a set of input variables $L$, *i.e.*, $\sum_{T:L\subseteq T\subseteq S}(-1)^{|T|-|L|}v_{\text{and}}(\boldsymbol{x}_L)$. Then, we compute all summations over the set $L \subseteq S$.

In this way, we can compute them separately for different cases of $L \subseteq T \subseteq S$. In the following, we consider the cases (1) $L = S = T$, and (2) $L \subseteq T \subseteq S, L \neq S$, respectively.

(1) When $L = S = T$, the linear combination of all subsets $T$ containing $L$ with respect to the model output $v_{\text{and}}(\boldsymbol{x}_L)$ is $(-1)^{|S|-|S|}v_{\text{and}}(\boldsymbol{x}_L) = v_{\text{and}}(\boldsymbol{x}_L)$.

(2) When $L \subseteq T \subseteq S, L \neq S$, the linear combination of all subsets $T$ containing $L$ with respect to the model output $v_{\text{and}}(\boldsymbol{x}_L)$ is $\sum_{T:L\subseteq T\subseteq S}(-1)^{|T|-|L|}v_{\text{and}}(\boldsymbol{x}_L)$. For all sets $T : S \supseteq T \supseteq L$, let us consider the linear combinations of all sets $T$ with number $|T|$ for the model output $v_{\text{and}}(\boldsymbol{x}_L)$, respectively. Let $m := |T|-|L|, (0 \leq m \leq |S|-|L|)$, then there are a total of $C_{|S|-|L|}^m$ combinations of all sets $T$ of order $|T|$. Thus, given $L$, accumulating the model outputs $v_{\text{and}}(\boldsymbol{x}_L)$ corresponding to all $T \supseteq L$, then $\sum_{T:L\subseteq T\subseteq S}(-1)^{|T|-|L|}v_{\text{and}}(\boldsymbol{x}_L) = v_{\text{and}}(\boldsymbol{x}_L) \cdot \underbrace{\sum_{m=0}^{|S|-|L|}C_{|S|-|L|}^m(-1)^m}_{=0} = 0$.

Please see the complete derivation of the following formula.

$$
\begin{aligned}
&\sum_{\emptyset\neq T\subseteq S}I_{\text{and}}(T|\boldsymbol{x})\\
=&\sum_{\emptyset\neq T\subseteq S}\sum_{L\subseteq T}(-1)^{|T|-|L|}v_{\text{and}}(\boldsymbol{x}_L)\\
=&\sum_{L\subseteq S}\sum_{T:L\subseteq T\subseteq S}(-1)^{|T|-|L|}v_{\text{and}}(\boldsymbol{x}_L)-v_{\text{and}}(\boldsymbol{x}_\emptyset)\\
=&\underbrace{v_{\text{and}}(\boldsymbol{x}_S)}_{L=S}+\sum_{L\subseteq S,L\neq S}v_{\text{and}}(\boldsymbol{x}_L)\cdot\underbrace{\sum_{m=0}^{|S|-|L|}C_{|S|-|L|}^m(-1)^m}_{=0}-v_{\text{and}}(\boldsymbol{x}_\emptyset)\\
=&v_{\text{and}}(\boldsymbol{x}_S)-v_{\text{and}}(\boldsymbol{x}_\emptyset)\\
=&v_{\text{and}}(\boldsymbol{x}_S)-v(\boldsymbol{x}_\emptyset)
\end{aligned}\tag{17}
$$

Thus, we have $\forall \emptyset \neq S \subseteq N, v_{\text{and}}(\boldsymbol{x}_S) = \sum_{\emptyset \neq T \subseteq S} I_{\text{and}}(T|\boldsymbol{x}) + v(\boldsymbol{x}_\emptyset)$.

**(2) Universal matching theorem of OR interactions.**

According to the definition of OR interactions, we will derive that $\forall S \subseteq N, v_{\text{or}}(\boldsymbol{x}_S) = \sum_{T:T\cap S\neq\emptyset} I_{\text{or}}(S|\boldsymbol{x})$, where we define $v_{\text{or}}(\boldsymbol{x}_\emptyset) = 0$ (recall that in Step (1), we attribute the output on empty input to AND interactions).

Specifically, the OR interaction is defined as $I_{\text{or}}(T|\boldsymbol{x}) = -\sum_{L\subseteq T}(-1)^{|T|-|L|}v_{\text{or}}(\boldsymbol{x}_{N\setminus L})$ in 2. Similar to the above derivation of the universal matching theorem of AND interactions, to compute the sum of OR interactions $\sum_{T:T\cap S\neq\emptyset} I_{\text{or}}(T|\boldsymbol{x}) = \sum_{T:T\cap S\neq\emptyset}\left[-\sum_{L\subseteq T}(-1)^{|T|-|L|}v_{\text{or}}(\boldsymbol{x}_{N\setminus L})\right]$, we first exchange the order of summation of the set $L \subseteq T \subseteq N$ and the set $T : T \cap S \neq \emptyset$. That is, we compute all linear combinations of all sets $T$ containing $L$ with respect to the model outputs $v_{\text{or}}(\boldsymbol{x}_{N\setminus L})$ given a set of input variables $L$, i.e., $\sum_{T:T\cap S\neq\emptyset, T\supseteq L}(-1)^{|T|-|L|}v_{\text{or}}(\boldsymbol{x}_{N\setminus L})$. Then, we compute all summations over the set $L \subseteq N$.

In this way, we can compute them separately for different cases of $L \subseteq T \subseteq N, T \cap S \neq \emptyset$. In the following, we consider the cases (1) $L = N \setminus S$, (2) $L = N$, (3) $L \cap S \neq \emptyset, L \neq N$, and (4) $L \cap S = \emptyset, L \neq N \setminus S$, respectively.

(1) When $L = N \setminus S$, the linear combination of all subsets $T$ containing $L$ with respect to the model output $v_{\text{or}}(\boldsymbol{x}_{N\setminus L})$ is $\sum_{T:T\cap S\neq\emptyset, T\supseteq L}(-1)^{|T|-|L|}v_{\text{or}}(\boldsymbol{x}_{N\setminus L}) = \sum_{T:T\cap S\neq\emptyset, T\supseteq L}(-1)^{|T|-|L|}v_{\text{or}}(\boldsymbol{x}_S)$. For all sets $T : T \supseteq L, T \cap S \neq \emptyset$ (then $T \neq N \setminus S, T \neq L$), let us consider the linear combinations of all sets $T$ with number $|T|$ for the model output $v_{\text{or}}(\boldsymbol{x}_S)$, respectively. Let $|T'| := |T| - |L|$, $(1 \leq |T'| \leq |S|)$, then there are a total of $C_{|S|}^{|T'|}$ combinations of all sets $T'$ of order $|T'|$. Thus, given $L$, accumulating the model outputs $v_{\text{or}}(\boldsymbol{x}_S)$ corresponding to all $T \supseteq L$, then

$$\sum_{T:T\cap S\neq\emptyset, T\supseteq L}(-1)^{|T|-|L|}v_{\text{or}}(\boldsymbol{x}_{N\setminus L}) = v_{\text{or}}(\boldsymbol{x}_S) \cdot \underbrace{\sum_{|T'|=1}^{|S|} C_{|S|}^{|T'|}(-1)^{|T'|}}_{=-1} = -v_{\text{or}}(\boldsymbol{x}_S).$$

(2) When $L = N$ (then $T = N$), the linear combination of all subsets $T$ containing $L$ with respect to the model output $v_{\text{or}}(\boldsymbol{x}_{N\setminus L})$ is $\sum_{T:T\cap S\neq\emptyset, T\supseteq L}(-1)^{|T|-|L|}v_{\text{or}}(\boldsymbol{x}_{N\setminus L}) = (-1)^{|N|-|N|}v_{\text{or}}(\boldsymbol{x}_\emptyset) = v_{\text{or}}(\boldsymbol{x}_\emptyset)$.

(3) When $L \cap S \neq \emptyset, L \neq N$, the linear combination of all subsets $T$ containing $L$ with respect to the model output $v_{\text{or}}(\boldsymbol{x}_{N\setminus L})$ is $\sum_{T:T\cap S\neq\emptyset, T\supseteq L}(-1)^{|T|-|L|}v_{\text{or}}(\boldsymbol{x}_{N\setminus L})$. For all sets $T : T \supseteq L, T \cap S \neq \emptyset$, let us consider the linear combinations of all sets $T$ with number $|T|$ for the model output $v_{\text{or}}(\boldsymbol{x}_S)$, respectively. Let us split $|T| - |L|$ into $|T'|$ and $|T''|$, i.e., $|T| - |L| = |T'| + |T''|$, where $T' = \{i|i \in T, i \notin L, i \in N \setminus S\}, T'' = \{i|i \in T, i \notin L, i \in S\}$ (then $0 \leq |T''| \leq |S| - |S \cap L|$) and $|T'| + |T''| + |L| = |T|$. In this way, there are a total of $C_{|S|-|S\cap L|}^{|T''|}$ combinations of all sets $T''$ of order $|T''|$. Thus, given $L$, accumulating the model outputs $v_{\text{or}}(\boldsymbol{x}_{N\setminus L})$ corresponding to all $T \supseteq L$, then $\sum_{T:T\cap S\neq\emptyset, T\supseteq L}(-1)^{|T|-|L|}v_{\text{or}}(\boldsymbol{x}_{N\setminus L}) =$

$$v_{\text{or}}(\boldsymbol{x}_{N\setminus L}) \cdot \sum_{T'\subseteq N\setminus S\setminus L} \underbrace{\sum_{|T''|=0}^{|S|-|S\cap L|} C_{|S|-|S\cap L|}^{|T''|}(-1)^{|T'|+|T''|}}_{=0} = 0.$$

(4) When $L \cap S = \emptyset, L \neq N \setminus S$, the linear combination of all subsets $T$ containing $L$ with respect to the model output $v_{\text{or}}(\boldsymbol{x}_{N\setminus L})$ is $\sum_{T:T\cap S\neq\emptyset, T\supseteq L}(-1)^{|T|-|L|}v_{\text{or}}(\boldsymbol{x}_{N\setminus L})$. Similarly, let us split $|T| - |L|$ into $|T'|$ and $|T''|$, i.e., $|T| - |L| = |T'| + |T''|$, where $T' = \{i|i \in T, i \notin L, i \in N \setminus S\}$, $T'' = \{i|i \in T, i \in S\}$ (then $0 \leq |T''| \leq |S|$) and $|T'| + |T''| + |L| = |T|$. In this way, there are a total of $C_{|S|}^{|T''|}$ combinations of all sets $T''$ of order $|T''|$. Thus, given $L$, accumulating the model outputs $v_{\text{or}}(\boldsymbol{x}_{N\setminus L})$ corresponding to all $T \supseteq L$, then $\sum_{T:T\cap S\neq\emptyset, T\supseteq L}(-1)^{|T|-|L|}v_{\text{or}}(\boldsymbol{x}_{N\setminus L}) =$

$$v_{\text{or}}(\boldsymbol{x}_{N\setminus L}) \cdot \sum_{T'\subseteq N\setminus S\setminus L} \underbrace{\sum_{|T''|=0}^{|S|} C_{|S|}^{|T''|}(-1)^{|T'|+|T''|}}_{=0} = 0.$$

Please see the complete derivation of the following formula.

$$
\begin{aligned}
\sum_{T:T\cap S\neq\emptyset} I_{\text{or}}(T|\boldsymbol{x}) &= \sum_{T:T\cap S\neq\emptyset}\left[-\sum_{L\subseteq T}(-1)^{|T|-|L|}v_{\text{or}}(\boldsymbol{x}_{N\setminus L})\right] \\
&= -\sum_{L\subseteq N}\sum_{T:T\cap S\neq\emptyset,T\supseteq L}(-1)^{|T|-|L|}v_{\text{or}}(\boldsymbol{x}_{N\setminus L}) \\
&= -\left[\sum_{|T'|=1}^{|S|}C_{|S|}^{|T'|}(-1)^{|T'|}\right]\cdot\underbrace{v_{\text{or}}(\boldsymbol{x}_S)}_{L=N\setminus S}-\underbrace{v_{\text{or}}(\boldsymbol{x}_\emptyset)}_{L=N} \\
&\quad -\sum_{L\cap S\neq\emptyset,L\neq N}\left[\sum_{T'\subseteq N\setminus S\setminus L}\left(\sum_{|T''|=0}^{|S|-|S\cap L|}C_{|S|-|S\cap L|}^{|T''|}(-1)^{|T'|+|T''|}\right)\right]\cdot v_{\text{or}}(\boldsymbol{x}_{N\setminus L}) \\
&\quad -\sum_{L\cap S=\emptyset,L\neq N\setminus S}\left[\sum_{T'\subseteq N\setminus S\setminus L}\left(\sum_{|T''|=0}^{|S|}C_{|S|}^{|T''|}(-1)^{|T'|+|T''|}\right)\right]\cdot v_{\text{or}}(\boldsymbol{x}_{N\setminus L}) \\
&= -(-1)\cdot v_{\text{or}}(\boldsymbol{x}_S)-v_{\text{or}}(\boldsymbol{x}_\emptyset)-\sum_{L\cap S\neq\emptyset,L\neq N}\left[\sum_{T'\subseteq N\setminus S\setminus L}0\right]\cdot v_{\text{or}}(\boldsymbol{x}_{N\setminus L}) \\
&\quad -\sum_{L\cap S=\emptyset,L\neq N\setminus S}\left[\sum_{T'\subseteq N\setminus S\setminus L}0\right]\cdot v_{\text{or}}(\boldsymbol{x}_{N\setminus L}) \\
&= v_{\text{or}}(\boldsymbol{x}_S)-v_{\text{or}}(\boldsymbol{x}_\emptyset) \\
&= v_{\text{or}}(\boldsymbol{x}_S)
\end{aligned}
\tag{18}
$$

### (3) Universal matching theorem of AND-OR interactions.

With the universal matching theorem of AND interactions and the universal matching theorem of OR interactions, we can easily get $v(\boldsymbol{x}_S) = v_{\text{and}}(\boldsymbol{x}_S) + v_{\text{or}}(\boldsymbol{x}_S) = v(\boldsymbol{x}_\emptyset) + \sum_{\emptyset\neq T\subseteq S}I_{\text{and}}(T|\boldsymbol{x}) + \sum_{T:T\cap S\neq\emptyset}I_{\text{or}}(T|\boldsymbol{x})$, thus, we obtain the universal matching theorem of AND-OR interactions.

$\square$

### F.2  Proof of Eq. (6) and Eq. (7)

Before we give the derivation of Eq. (6) and Eq. (7), we first prove the following lemma.

**Lemma 3.** *The effect $I(T|\boldsymbol{x})$ of an AND interaction w.r.t. subset $T$ on sample $\boldsymbol{x}$ can be rewritten as*

$$
I(T|\boldsymbol{x}) = \sum_{\boldsymbol{\pi}\in Q_T}\frac{1}{\prod_{i=1}^{n}\pi_i!}\frac{\partial^{\pi_1+\cdots+\pi_n}v}{\partial x_1^{\pi_1}\cdots\partial x_n^{\pi_n}}\bigg|_{\boldsymbol{x}=\boldsymbol{x}_\emptyset}\prod_{i\in T}(x_i-b_i)^{\pi_i},
\tag{19}
$$

*where $Q_T = \{[\pi_1,\ldots,\pi_n]^\top\mid\forall i\in T,\pi_i\in\mathbb{N}^+;\forall i\notin T,\pi_i=0\}$.*

Note that a similar proof was first introduced in [26].

*Proof.* Let us denote the function on the right of Eq. (19) by $K(T|\boldsymbol{x})$, *i.e.*, for $S\neq\emptyset$,

$$
K(T|\boldsymbol{x}) \stackrel{\text{def}}{=} \sum_{\boldsymbol{\pi}\in Q_T}\frac{1}{\prod_{i=1}^{n}\pi_i!}\frac{\partial^{\pi_1+\cdots+\pi_n}v}{\partial x_1^{\pi_1}\cdots\partial x_n^{\pi_n}}\bigg|_{\boldsymbol{x}=\boldsymbol{x}_\emptyset}\prod_{i\in T}(x_i-b_i)^{\pi_i}.
\tag{20}
$$

Actually, it has been proven in [13] and [23] that the AND interaction $I(T|\boldsymbol{x})$ (see definition in Eq. (2)) is the **unique** metric satisfying the following property (an extension of the property for AND-OR interactions is mentioned in Theorem 2), *i.e.*,

$$
\forall S\subseteq N,\ v(\boldsymbol{x}_S) = \sum_{\emptyset\neq T\subseteq S}I(T|\boldsymbol{x})+v(\boldsymbol{x}_\emptyset).
\tag{21}
$$

Thus, as long as we can prove that $K(T|\boldsymbol{x})$ also satisfies the above universal matching property, we can obtain $I(T|\boldsymbol{x}) = K(T|\boldsymbol{x})$.

To this end, we only need to prove $K(T|\boldsymbol{x})$ also satisfies the property in Eq. (21). Specifically, given an input sample $\boldsymbol{x} \in \mathbb{R}^n$, let us consider the Taylor expansion of the network output $v(\boldsymbol{x}_S)$ of an arbitrarily masked sample $\boldsymbol{x}_S$, which is expanded at $\boldsymbol{x}_\emptyset = \boldsymbol{b} = [b_1, \cdots, b_n]^\top$. Then, we have

$$\forall S \subseteq N, \ v(\boldsymbol{x}_S) = \sum_{\pi_1=0}^{\infty} \cdots \sum_{\pi_n=0}^{\infty} \frac{1}{\prod_{i=1}^n \pi_i!} \frac{\partial^{\pi_1 + \cdots + \pi_n} v}{\partial x_1^{\pi_1} \cdots \partial x_n^{\pi_n}} \bigg|_{\boldsymbol{x}=\boldsymbol{x}_\emptyset} \prod_{i=1}^n ((\boldsymbol{x}_S)_i - b_i)^{\pi_i} \tag{22}$$

where $b_i$ denotes the baseline value to mask the input variable $x_i$.

According to the definition of the masked sample $\boldsymbol{x}_S$, we have that all variables in $S$ keep unchanged and other variables are masked to the baseline value. That is, $\forall i \in S, (\boldsymbol{x}_S)_i = x_i; \forall i \notin S, (\boldsymbol{x}_S)_i = b_i$. Hence, we obtain $\forall i \notin S, ((\boldsymbol{x}_S)_i - b_i)^{\pi_i} = 0$ if $\pi_i > 0$. Then, among all Taylor expansion terms, only terms corresponding to degrees $\boldsymbol{\pi}$ in the set $P_S = \{[\pi_1, \cdots, \pi_n]^\top \mid \forall i \in S, \pi_i \in \mathbb{N}; \forall i \notin S, \pi_i = 0\}$ may not be zero (we consider the value of $((\boldsymbol{x}_S)_i - b_i)^{\pi_i}$ to be always equal to 1 if $\pi_i = 0$). Therefore, Eq. (22) can be re-written as

$$\forall S \subseteq N, \quad v(\boldsymbol{x}_S) = \sum_{\boldsymbol{\pi} \in P_S} \frac{1}{\prod_{i=1}^n \pi_i!} \frac{\partial^{\pi_1 + \cdots + \pi_n} v}{\partial x_1^{\pi_1} \cdots \partial x_n^{\pi_n}} \bigg|_{\boldsymbol{x}=\boldsymbol{x}_\emptyset} \prod_{i \in S} (x_i - b_i)^{\pi_i}. \tag{23}$$

We find that the set $P_S$ can be divided into multiple disjoint sets as $P_S = \cup_{T \subseteq S} Q_T$, where $Q_T = \{[\pi_1, \cdots, \pi_n]^\top \mid \forall i \in T, \pi_i \in \mathbb{N}^+; \forall i \notin T, \pi_i = 0\}$. Then, we can further write Eq. (23) as

$$\forall S \subseteq N, \quad v(\boldsymbol{x}_S) = \sum_{T \subseteq S} \sum_{\boldsymbol{\pi} \in Q_T} \frac{1}{\prod_{i=1}^n \pi_i!} \frac{\partial^{\pi_1 + \cdots + \pi_n} v}{\partial x_1^{\pi_1} \cdots \partial x_n^{\pi_n}} \bigg|_{\boldsymbol{x}=\boldsymbol{x}_\emptyset} \prod_{i \in T} (x_i - b_i)^{\pi_i}$$

$$= \sum_{\emptyset \neq T \subseteq S} K(T|\boldsymbol{x}) + v(\boldsymbol{x}_\emptyset). \quad \text{// according to the definition of } K(T|\boldsymbol{x}) \text{ in Eq. (20)}$$

$$\tag{24}$$

The last step is obtained as follows. When $T = \emptyset$, $Q_T$ only has one element $\boldsymbol{\pi} = [0, \cdots, 0]^\top$, which corresponds to the term $v(\boldsymbol{x}_\emptyset)$.

Thus, $K(T|\boldsymbol{x})$ satisfies the property in Eq. (21), and this means $I(T|\boldsymbol{x}) = K(T|\boldsymbol{x}) = \sum_{\boldsymbol{\pi} \in Q_T} \frac{1}{\prod_{i=1}^n \pi_i!} \frac{\partial^{\pi_1 + \cdots + \pi_n} v}{\partial x_1^{\pi_1} \cdots \partial x_n^{\pi_n}} \big|_{\boldsymbol{x}=\boldsymbol{x}_\emptyset} \prod_{i \in T} (x_i - b_i)^{\pi_i}$.

$\square$

Then, let us continue the proof of Eq. (6) and Eq. (7).

*Proof.* Given a specific sample $\hat{\boldsymbol{x}}$, let us consider the following function defined in Eq. (6) and Eq. (7).

$$f(\boldsymbol{x}) = \sum_{T \subseteq N} w_T \, J_T(\boldsymbol{x}), \tag{25}$$

where the scalar weight $w_T = I(T|\boldsymbol{x} = \hat{\boldsymbol{x}})$, and the function $J_T(\boldsymbol{x}) = \sum_{\boldsymbol{\pi} \in Q_T} \frac{1}{\prod_{i=1}^n \pi_i!} \frac{\partial^{\pi_1 + \cdots + \pi_n} v}{\partial x_1^{\pi_1} \cdots \partial x_n^{\pi_n}} \big|_{\boldsymbol{x}=\boldsymbol{x}_\emptyset} \prod_{i \in T} (x_i - b_i)^{\pi_i} / w_T$.

We will then prove that $\forall S \subseteq N, \ f(\hat{\boldsymbol{x}}_S) = v(\hat{\boldsymbol{x}}_S)$.

$$f(\hat{\boldsymbol{x}}_S) = \sum_{T \subseteq N} w_T \, J_T(\hat{\boldsymbol{x}}_S) \tag{26}$$

$$= \sum_{T \subseteq N} \sum_{\boldsymbol{\pi} \in Q_T} \frac{1}{\prod_{i=1}^{n} \pi_i!} \frac{\partial^{\pi_1 + \cdots + \pi_n} v}{\partial x_1^{\pi_1} \cdots \partial x_n^{\pi_n}} \bigg|_{\boldsymbol{x}=\boldsymbol{x}_\emptyset} \prod_{i \in T} ((\hat{\boldsymbol{x}}_S)_i - b_i)^{\pi_i} \ \text{// } w_T \text{ cancels out} \tag{27}$$

$$= \sum_{T \subseteq S} \sum_{\boldsymbol{\pi} \in Q_T} \frac{1}{\prod_{i=1}^{n} \pi_i!} \frac{\partial^{\pi_1 + \cdots + \pi_n} v}{\partial x_1^{\pi_1} \cdots \partial x_n^{\pi_n}} \bigg|_{\boldsymbol{x}=\boldsymbol{x}_\emptyset} \prod_{i \in T} ((\hat{\boldsymbol{x}}_S)_i - b_i)^{\pi_i} \tag{28}$$

$\quad$ // if $T \nsubseteq S$, then $\exists j \in T \setminus S$, s.t. $(\hat{\boldsymbol{x}}_S)_j - b_j = 0$, which makes the whole term zero

$$\tag{29}$$

$$= \sum_{T \subseteq S} \sum_{\boldsymbol{\pi} \in Q_T} \frac{1}{\prod_{i=1}^{n} \pi_i!} \frac{\partial^{\pi_1 + \cdots + \pi_n} v}{\partial x_1^{\pi_1} \cdots \partial x_n^{\pi_n}} \bigg|_{\boldsymbol{x}=\boldsymbol{x}_\emptyset} \prod_{i \in T} (\hat{x}_i - b_i)^{\pi_i} \tag{30}$$

$\quad$ // when $T \subseteq S$, we have $\forall i \in T, (\hat{\boldsymbol{x}}_S)_i = \hat{x}_i$ $\tag{31}$

$$= \sum_{\emptyset \neq T \subseteq S} I(T|\boldsymbol{x} = \hat{\boldsymbol{x}}) + v(\boldsymbol{x}_\emptyset) \ \text{// the inverse direction of Lemma 3 we have just proven}$$

$$\tag{32}$$

$$= v(\hat{\boldsymbol{x}}_S) \ \text{// the inverse direction of universal matching theorem} \tag{33}$$

$\square$

*Remark.* The function $f(\boldsymbol{x})$ essentially provides a continuous implementation of Eq. (3) in the universal matching theorem (Theorem 2). The weight $w_T = I(T|\boldsymbol{x} = \hat{\boldsymbol{x}})$ is the interaction effect *w.r.t.* to subset $T$ on the *unmasked* sample $\hat{\boldsymbol{x}}$, while the function $J_T(\boldsymbol{x})$ is a continuous extension of the indicator function $\mathbb{1}(\hat{\boldsymbol{x}}_S$ triggers the AND relation $T)$ (thus we call $J_T(\boldsymbol{x})$ a *triggering function* and the value of this function *triggering strength*).

### F.3 Proof of Lemma 1

*Proof.* Given the inference scores on masked samples $\{\widetilde{v}(\boldsymbol{x}_S) : S \subseteq N\}$, the interaction between input variables *w.r.t.* $T \subseteq N$ can be computed as $\widetilde{I}(T|\boldsymbol{x}) = \sum_{S \subseteq T}(-1)^{|T|-|S|} \widetilde{v}(\boldsymbol{x}_S)$ (the computation of AND interactions in Eq. (2)).

Since we assume that $\forall S \subseteq N, \widetilde{v}(\boldsymbol{x}_S) = v(\boldsymbol{x}_S) + \Delta v_S, \Delta v_S \sim \mathcal{N}(0, \sigma^2), \widetilde{I}(T|\boldsymbol{x})$ can be written as

$$\widetilde{I}(T|\boldsymbol{x}) = \sum_{S \subseteq T}(-1)^{|T|-|S|} \widetilde{v}(\boldsymbol{x}_S) \tag{34}$$

$$= \sum_{S \subseteq T}(-1)^{|T|-|S|} (v(\boldsymbol{x}_S) + \Delta v_S) \tag{35}$$

$$= \sum_{S \subseteq T}(-1)^{|T|-|S|} v(\boldsymbol{x}_S) + \sum_{S \subseteq T}(-1)^{|T|-|S|}\Delta v_S \tag{36}$$

$$= I(T|\boldsymbol{x}) + \Delta I_T \tag{37}$$

where $I(T|\boldsymbol{x}) = \sum_{S \subseteq T}(-1)^{|T|-|S|}v(\boldsymbol{x}_S)$ is a noiseless component (not a random variable), and $\Delta I_T = \sum_{S \subseteq T}(-1)^{|T|-|S|}\Delta v_S$ is the noise component on the interaction.

Since each Gaussian noise $\Delta v_S \sim \mathcal{N}(0, \sigma^2), \forall S \subseteq N$, is independent and identically distributed, it is easy to see $\mathbb{E}[\Delta I_T] = \sum_{S \subseteq T}(-1)^{|T|-|S|}\mathbb{E}[\Delta v_S] = 0$. The variance of $\Delta I_T$ is computed as

$$\text{Var}[\Delta I_T] = \text{Var}(\sum_{S \subseteq T}(-1)^{|T|-|S|}\Delta v_S) \tag{38}$$

$$= \text{Var}(\Delta v_{S_1}) + \text{Var}(\Delta v_{S_2}) + \cdots + \text{Var}(\Delta v_{S_{2^{|T|}}}) \tag{39}$$

$$= 2^{|T|} \cdot \sigma^2, \tag{40}$$

because there are a total of $2^{|T|}$ subsets for $S \subseteq T$.

Furthermore, according to the analytic form of interaction effect in Eq. (19), we note that the values of $\widetilde{I}(T|\boldsymbol{x})$ and $\widetilde{J}_T(\boldsymbol{x})$ have a ratio of $w_T$. Therefore, if we write $\widetilde{J}_T(\boldsymbol{x}) = J_T(\boldsymbol{x}) + \epsilon_T$, then the noise term satisfies $\epsilon_T = \Delta I_T / w_T$, and thus $\mathbb{E}[\epsilon_T] = 0, \mathrm{Var}[\epsilon_T] \propto 2^{|T|}\sigma^2$.

$\square$

### F.4 Proof of Theorem 3

*Proof.* We concatenate all $\boldsymbol{J}(\boldsymbol{x}_S)$ (*w.r.t.* all $2^n$ masked samples $\boldsymbol{x}_S$, $S \subseteq N$) into a matrix $\boldsymbol{J} = [\boldsymbol{J}(\boldsymbol{x}_{S_1}), \boldsymbol{J}(\boldsymbol{x}_{S_2}), \cdots, \boldsymbol{J}(\boldsymbol{x}_{S_{2^n}})]^\top \in \{0,1\}^{2^n \times 2^n}$ to represent the triggering strength of $2^n$ interactions on $2^n$ masked samples We also concatenate all noise terms on all $2^n$ masked samples into a matrix $\boldsymbol{\mathcal{E}} = [\boldsymbol{\epsilon}^{(1)}, \boldsymbol{\epsilon}^{(2)}, \cdots, \boldsymbol{\epsilon}^{(2^n)}]^\top$ to represent the noise term over $\boldsymbol{J}$. We concatenate the output score vector $\boldsymbol{y} \overset{\text{def}}{=} [y(\boldsymbol{x}_{S_1}), y(\boldsymbol{x}_{S_2}), \cdots, y(\boldsymbol{x}_{S_{2^n}})]^\top \in \mathbb{R}^{2^n}$ to represent the finally converged outputs on all $2^n$ masked samples.

The optimal weights $\hat{\boldsymbol{w}}$ can be solved by minimizing the loss function $\widetilde{L}(\boldsymbol{w})$ in Eq. (9). The loss function can be rewritten as follows:

$$\hat{\boldsymbol{w}} = \arg\min_{\boldsymbol{w}} \widetilde{L}(\boldsymbol{w}) \tag{41}$$

$$\widetilde{L}(\boldsymbol{w}) = \mathbb{E}_{\boldsymbol{\epsilon}}\mathbb{E}_{S \subseteq N}\left[\left(y_S - \boldsymbol{w}^\top(\boldsymbol{J}(\boldsymbol{x}_S) + \boldsymbol{\epsilon})\right)^2\right], \tag{42}$$

$$= \mathbb{E}_{\boldsymbol{\mathcal{E}}}\left[\frac{1}{2^n}\|\boldsymbol{y} - (\boldsymbol{J} + \boldsymbol{\mathcal{E}})\boldsymbol{w}\|_2^2\right], \tag{43}$$

$$= \frac{1}{2^n}\mathbb{E}_{\boldsymbol{\mathcal{E}}}\left[(\boldsymbol{y} - (\boldsymbol{J} + \boldsymbol{\mathcal{E}})\boldsymbol{w})^\top(\boldsymbol{y} - (\boldsymbol{J} + \boldsymbol{\mathcal{E}})\boldsymbol{w})\right], \tag{44}$$

$$= \frac{1}{2^n}\left(\boldsymbol{y}^\top\boldsymbol{y} - 2\boldsymbol{y}^\top\mathbb{E}_{\boldsymbol{\mathcal{E}}}[(\boldsymbol{J} + \boldsymbol{\mathcal{E}})]\boldsymbol{w} + \boldsymbol{w}^\top\mathbb{E}_{\boldsymbol{\mathcal{E}}}\left[(\boldsymbol{J} + \boldsymbol{\mathcal{E}})^\top(\boldsymbol{J} + \boldsymbol{\mathcal{E}})\right]\boldsymbol{w}\right). \tag{45}$$

Taking the derivative with respect to $\boldsymbol{w}$ and setting it to zero, we get:

$$\frac{\partial \widetilde{L}}{\partial \boldsymbol{w}} = -2\mathbb{E}_{\boldsymbol{\mathcal{E}}}\left[(\boldsymbol{J} + \boldsymbol{\mathcal{E}})^\top\boldsymbol{y}\right] + 2\mathbb{E}_{\boldsymbol{\mathcal{E}}}\left[(\boldsymbol{J} + \boldsymbol{\mathcal{E}})^\top(\boldsymbol{J} + \boldsymbol{\mathcal{E}})\boldsymbol{w}\right] = 0, \tag{46}$$

$$\Rightarrow \mathbb{E}_{\boldsymbol{\mathcal{E}}}\left[(\boldsymbol{J} + \boldsymbol{\mathcal{E}})^\top(\boldsymbol{J} + \boldsymbol{\mathcal{E}})\right]\boldsymbol{w} = \mathbb{E}_{\boldsymbol{\mathcal{E}}}\left[(\boldsymbol{J} + \boldsymbol{\mathcal{E}})^\top\boldsymbol{y}\right], \tag{47}$$

$$\Rightarrow (\boldsymbol{J}^\top\boldsymbol{J} + \mathbb{E}_{\boldsymbol{\mathcal{E}}}[\boldsymbol{\mathcal{E}}^\top\boldsymbol{J}] + \boldsymbol{J}^\top\mathbb{E}_{\boldsymbol{\mathcal{E}}}[\boldsymbol{\mathcal{E}}] + \mathbb{E}_{\boldsymbol{\mathcal{E}}}[\boldsymbol{\mathcal{E}}^\top\boldsymbol{\mathcal{E}}])\boldsymbol{w} = \boldsymbol{J}^\top\boldsymbol{y}, \tag{48}$$

$$\Rightarrow (\boldsymbol{J}^\top\boldsymbol{J} + \mathbb{E}_{\boldsymbol{\mathcal{E}}}[\boldsymbol{\mathcal{E}}^\top\boldsymbol{\mathcal{E}}])\boldsymbol{w} = \boldsymbol{J}^\top\boldsymbol{y}. \quad \text{// because } \mathbb{E}[\boldsymbol{\mathcal{E}}] = \boldsymbol{0} \tag{49}$$

Notice that the sample covariance matrix $\frac{1}{m}\boldsymbol{\mathcal{E}}^\top\boldsymbol{\mathcal{E}}$ converges to the true covariance matrix $\mathrm{Cov}(\boldsymbol{\mathcal{E}})$, when $m = 2^n$ is large. Therefore, $\mathbb{E}_{\boldsymbol{\mathcal{E}}}[\boldsymbol{\mathcal{E}}^\top\boldsymbol{\mathcal{E}}]) = \mathbb{E}_{\boldsymbol{\mathcal{E}}}[2^n\mathrm{Cov}(\boldsymbol{\mathcal{E}})] = 2^n\mathrm{Cov}(\boldsymbol{\mathcal{E}})$. Because we assume noises on different interactions are independent, it is a diagonal matrix, denoted by $\mathrm{Cov}(\boldsymbol{\mathcal{E}}) = \mathrm{diag}(\boldsymbol{c})$, where $\boldsymbol{c} = \mathrm{vec}(\{\mathrm{Var}[\epsilon_T] : T \subseteq N\}) = \mathrm{vec}(\{2^{|T|}\sigma^2 : T \subseteq N\}) \in \mathbb{R}^{2^n}$ denotes the vector of variances of the triggering strength of $2^n$ interactions.

Thus, we have:

$$(\boldsymbol{J}^\top\boldsymbol{J} + 2^n\mathrm{diag}(\boldsymbol{c}))\boldsymbol{w} = \boldsymbol{J}^\top\boldsymbol{y}. \tag{50}$$

Next, we can prove that the matrix $\boldsymbol{J}^\top\boldsymbol{J} + 2^n\mathrm{diag}(\boldsymbol{c})$ is always invertible, as follows. (1) We can prove that $\boldsymbol{J}^\top\boldsymbol{J}$ is positive semi-definite, because $\forall \boldsymbol{u} \neq \boldsymbol{0}, \boldsymbol{u}^\top\boldsymbol{J}^\top\boldsymbol{J}\boldsymbol{u} = \|\boldsymbol{J}\boldsymbol{u}\|_2^2 \geq 0$. (2) We can further prove that $\boldsymbol{J}^\top\boldsymbol{J}$ is positive definite. Let us denote the eigenvalues of $\boldsymbol{J}^\top\boldsymbol{J}$ as $\lambda_1, \cdots, \lambda_{2^n} \in \mathbb{R}$ (because $\boldsymbol{J}^\top\boldsymbol{J}$ is real symmetric, its eigenvalues must be real). Note that the diagonal elements of $\boldsymbol{J}^\top\boldsymbol{J}$ are all positive, so we have $\prod_{i=1}^{2^n} \lambda_i = \prod_{i=1}^{2^n}(\boldsymbol{J}^\top\boldsymbol{J})_{ii} > 0$. Combining the positive semi-definiteness, we know that the eigenvalues of $\boldsymbol{J}^\top\boldsymbol{J}$ must be all positive, without having a zero eigenvalue. It means that $\boldsymbol{J}^\top\boldsymbol{J}$ is positive definite. (3) We can prove that $\boldsymbol{J}^\top\boldsymbol{J} + 2^n\mathrm{diag}(\boldsymbol{c})$ is positive definite. The diagonal matrix $2^n\mathrm{diag}(\boldsymbol{c})$ is positive definite, because all its diagonal elements are positive. The sum of two positive definite matrices is still positive definite. (4) Since $\boldsymbol{J}^\top\boldsymbol{J} + 2^n\mathrm{diag}(\boldsymbol{c})$ is positive definite, it cannot have a zero eigenvalue, and is thus invertible.

So the optimal weights can be solved as

$$\hat{\boldsymbol{w}} = (\boldsymbol{J}^\top \boldsymbol{J} + 2^n \mathrm{diag}(\boldsymbol{c}))^{-1} \boldsymbol{J}^\top \boldsymbol{y}. \tag{51}$$

Next we will show that $\boldsymbol{y} = \boldsymbol{J}^\top \boldsymbol{w}^*$. Recall that definition of $y(\boldsymbol{x}_S)$ is given by $y(\boldsymbol{x}_S) = v(\boldsymbol{x}_\emptyset) + \sum_{\emptyset \neq T \subseteq S} w_T^*$ in the main paper. According to the Lemma 2, we have $J_T(\boldsymbol{x}) = \mathbb{1}(T \subseteq S)$. Therefore, $y(\boldsymbol{x}_S)$ can be rewritten as $y(\boldsymbol{x}_S) = \sum_{T \subseteq N} J_T(\boldsymbol{x}_S) w_T^*$, where we define $w_\emptyset^* \overset{\text{def}}{=} v(\boldsymbol{x}_\emptyset)$ for simplicity of notation. Writing the sum in vector norm, we obtain $y(\boldsymbol{x}_S) = \boldsymbol{J}(\boldsymbol{x}_S)^\top \boldsymbol{w}^*$. Furthermore, the whole vector $\boldsymbol{y}$ can be written as $\boldsymbol{y} = \boldsymbol{J}^\top \boldsymbol{w}^*$.

With $\boldsymbol{y} = \boldsymbol{J}^\top \boldsymbol{w}^*$, we have $\hat{\boldsymbol{w}} = (\boldsymbol{J}^\top \boldsymbol{J} + 2^n \mathrm{diag}(\boldsymbol{c}))^{-1} \boldsymbol{J}^\top \boldsymbol{J} \boldsymbol{w}^* = \hat{\boldsymbol{M}} \boldsymbol{w}^*$. $\qquad\square$

### F.5 Proof of Lemma 2

*Proof.* According to Eq. (7), the interaction triggering function on an arbitrarily given sample $\hat{\boldsymbol{x}}$ is given by

$$J_T(\boldsymbol{x}) = \sum_{\boldsymbol{\pi} \in Q_T} \frac{1}{\prod_{i=1}^n \pi_i!} \frac{\partial^{\pi_1 + \cdots + \pi_n} v}{\partial x_1^{\pi_1} \cdots \partial x_n^{\pi_n}} \Big|_{\boldsymbol{x} = \boldsymbol{x}_\emptyset} \prod_{i \in T} (x_i - b_i)^{\pi_i} / w_T \tag{52}$$

where $w_T = I(T | \boldsymbol{x} = \hat{\boldsymbol{x}})$, and $Q_T = \{[\pi_1, \ldots, \pi_n]^\top : \forall i \in T, \pi_i \in \mathbb{N}^+; \forall i \notin T, \pi_i = 0\}$.

Specifically, now we consider a masked sample $\hat{\boldsymbol{x}}_S$, and we will prove that $J_T(\hat{\boldsymbol{x}}_S) = \mathbb{1}(T \subseteq S)$. We consider the following two cases.

**Case 1:** $T \nsubseteq S$. Then, there exists some $j \in T \setminus S$. Since $j \notin S$, according to the masking rule of the sample $\hat{\boldsymbol{x}}_S$, we have $(\hat{\boldsymbol{x}}_S)_j - b_j = 0$. Since $j \in T$, we have $\pi_j \in \mathbb{N}^+$. Therefore, $((\hat{\boldsymbol{x}}_S)_j - b_j)^{\pi_j} = 0$. In this way, we have

$$\forall \boldsymbol{\pi} \in Q_T, \quad \prod_{i \in T} ((\hat{\boldsymbol{x}}_S)_i - b_i)^{\pi_i} = 0. \tag{53}$$

Since each term in the summation equals zero, we have $J_T(\hat{\boldsymbol{x}}_S) = 0$.

**Case 2:** $T \subseteq S$. In this case, $\forall i \in T$, we have $i \in S$. Therefore, according to the masking rule, we have $\forall i \in T \Rightarrow i \in S \Rightarrow (\hat{\boldsymbol{x}}_S)_i = \hat{x}_i$.

According to the analytic form of $I(T | \boldsymbol{x})$ in Eq. (19) in the proof in Appendix F.2, we can derive the value of $w_T$ as

$$w_T = I(T | \boldsymbol{x} = \hat{\boldsymbol{x}}) = \sum_{\boldsymbol{\pi} \in Q_T} \frac{1}{\prod_{i=1}^n \pi_i!} \frac{\partial^{\pi_1 + \cdots + \pi_n} v}{\partial x_1^{\pi_1} \cdots \partial x_n^{\pi_n}} \Big|_{\boldsymbol{x} = \boldsymbol{x}_\emptyset} \prod_{i \in T} (\hat{x}_i - b_i)^{\pi_i}. \tag{54}$$

Therefore, we can derive the value of $J_T(\hat{\boldsymbol{x}}_S)$ as follows.

$$J_T(\hat{\boldsymbol{x}}_S) = \sum_{\boldsymbol{\pi} \in Q_T} \frac{1}{\prod_{i=1}^n \pi_i!} \frac{\partial^{\pi_1 + \cdots + \pi_n} v}{\partial x_1^{\pi_1} \cdots \partial x_n^{\pi_n}} \Big|_{\boldsymbol{x} = \boldsymbol{x}_\emptyset} \prod_{i \in T} ((\hat{\boldsymbol{x}}_S)_i - b_i)^{\pi_i} / w_T \tag{55}$$

$$= \frac{\sum_{\boldsymbol{\pi} \in Q_T} \frac{1}{\prod_{i=1}^n \pi_i!} \frac{\partial^{\pi_1 + \cdots + \pi_n} v}{\partial x_1^{\pi_1} \cdots \partial x_n^{\pi_n}} \big|_{\boldsymbol{x} = \boldsymbol{x}_\emptyset} \prod_{i \in T} ((\hat{\boldsymbol{x}}_S)_i - b_i)^{\pi_i}}{\sum_{\boldsymbol{\pi} \in Q_T} \frac{1}{\prod_{i=1}^n \pi_i!} \frac{\partial^{\pi_1 + \cdots + \pi_n} v}{\partial x_1^{\pi_1} \cdots \partial x_n^{\pi_n}} \big|_{\boldsymbol{x} = \boldsymbol{x}_\emptyset} \prod_{i \in T} (\hat{x}_i - b_i)^{\pi_i}} \quad \text{// by Eq. (54)} \tag{56}$$

$$= \frac{\sum_{\boldsymbol{\pi} \in Q_T} \frac{1}{\prod_{i=1}^n \pi_i!} \frac{\partial^{\pi_1 + \cdots + \pi_n} v}{\partial x_1^{\pi_1} \cdots \partial x_n^{\pi_n}} \big|_{\boldsymbol{x} = \boldsymbol{x}_\emptyset} \prod_{i \in T} (\hat{x}_i - b_i)^{\pi_i}}{\sum_{\boldsymbol{\pi} \in Q_T} \frac{1}{\prod_{i=1}^n \pi_i!} \frac{\partial^{\pi_1 + \cdots + \pi_n} v}{\partial x_1^{\pi_1} \cdots \partial x_n^{\pi_n}} \big|_{\boldsymbol{x} = \boldsymbol{x}_\emptyset} \prod_{i \in T} (\hat{x}_i - b_i)^{\pi_i}} \tag{57}$$

$$\text{// because we have proven } \forall i \in T, \ (\hat{\boldsymbol{x}}_S)_i = \hat{x}_i \tag{58}$$

$$= 1 \tag{59}$$

Combining the two cases, we can conclude that $J_T(\hat{\boldsymbol{x}}_S) = \mathbb{1}(T \subseteq S)$.

In this way, no matter how we change the DNN $v(\cdot)$ or the input sample $\boldsymbol{x}$, the matrix $\boldsymbol{J} = [\boldsymbol{J}(\boldsymbol{x}_{S_1}), \boldsymbol{J}(\boldsymbol{x}_{S_2}), \cdots, \boldsymbol{J}(\boldsymbol{x}_{S_{2^n}})]^\top \in \{0, 1\}^{2^n \times 2^n}$ in Eq. (10) is a always a fixed binary matrix.

$\qquad\square$

### F.6 Proof of Theorem 4

*Proof.* We prove that for any two subsets $T, T' \subseteq N$ of the same order, the vector $\hat{\boldsymbol{m}}_T$ is a permutation of the vector $\hat{\boldsymbol{m}}_{T'}$.

The proof consists of two steps. First, we show that there exists a symmetric matrix transformation $\mathcal{T}(\cdot) = P_k P_{k-1} \cdots P_1 (\cdot) P_1 P_2 \cdots P_{k-1} P_k$, where $P_i$ is a permutation matrix, that maps both $\boldsymbol{J}^\top \boldsymbol{J}$ and $\boldsymbol{J}^\top \boldsymbol{J} + 2^n \mathrm{diag}(\boldsymbol{c})$ to themselves, i.e., $\mathcal{T}(\boldsymbol{J}^\top \boldsymbol{J}) = \boldsymbol{J}^\top \boldsymbol{J}$, $\mathcal{T}(\boldsymbol{J}^\top \boldsymbol{J} + 2^n \mathrm{diag}(\boldsymbol{c})) = \boldsymbol{J}^\top \boldsymbol{J} + 2^n \mathrm{diag}(\boldsymbol{c})$. We will show that this transformation $\mathcal{T}(\cdot)$ applies permutation to the rows and columns **of the same order**.

Second, we show that this transformation also maps $\hat{\boldsymbol{M}}$ to itself, i.e., $\mathcal{T}(\hat{\boldsymbol{M}}) = \hat{\boldsymbol{M}}$, implying that row vectors of the same order in $\hat{\boldsymbol{M}}$ are permutations of each other.

From Theorem 3, we have:

$$(\boldsymbol{J}^\top \boldsymbol{J} + 2^n \mathrm{diag}(\boldsymbol{c}))\hat{\boldsymbol{M}} = \boldsymbol{J}^\top \boldsymbol{J} \tag{60}$$

To simplify the notation, we denote $\boldsymbol{B} := \boldsymbol{J}^\top \boldsymbol{J}$ and $\boldsymbol{D} := 2^n \mathrm{diag}(\boldsymbol{c})$. Then, we have:

$$(\boldsymbol{B} + \boldsymbol{D})\hat{\boldsymbol{M}} = \boldsymbol{B} \tag{61}$$

**Step 1:** We construct a transformation $\mathcal{T}(\cdot)$ which permutes the rows and columns of a $2^n \times 2^n$ matrix based on element selection. Let us first consider the matrix $\boldsymbol{B}$. For the matrix $\boldsymbol{D}$, the analysis is similar because its diagonal elements $2^{|T|+n}\sigma^2$ are the same for each order. Thus, if $\mathcal{T}(\cdot)$ maps $\boldsymbol{B}$ to itself, it also maps $\boldsymbol{D}$ to itself.

Given the set $N = \{1, 2, \cdots, n\}$, the subsets $S_1, S_2, \cdots, S_{2^n}$ can be regarded as selections from the power set of $N$, denoted as $2^N$. Consider a permutation $\mathcal{P}$ acting on $N$. Under this permutation, the selections $S_1, S_2, \cdots, S_{2^n}$ transform correspondingly. For example, if $N = \{1, 2, 3\}$ is mapped to $N = \{3, 2, 1\}$ under the permutation $\mathcal{P}$, the list of subsets $[S_1, S_2, \cdots, S_{2^n}] = [\emptyset, \{1\}, \{2\}, \{3\}, \{1, 2\}, \{1, 3\}, \{2, 3\}, \{1, 2, 3\}]$ is mapped to $[\emptyset, \{3\}, \{2\}, \{1\}, \{3, 2\}, \{3, 1\}, \{2, 1\}, \{3, 2, 1\}]$.

This permutation induces a transformation $\mathcal{T}(\cdot) = P_k P_{k-1} \cdots P_1 (\cdot) P_1 P_2 \cdots P_{k-1} P_k$ on the matrix $\boldsymbol{B} = \boldsymbol{J}^\top \boldsymbol{J}$ by permuting its rows and columns.

Since the permutation acts on $N$ and preserves the inclusion relation, the transformation $\mathcal{T}(\cdot)$ is invariant, meaning $\mathcal{T}(\boldsymbol{B}) = \boldsymbol{B}$. Similarly, we have $\mathcal{T}(\boldsymbol{B} + \boldsymbol{D}) = \boldsymbol{B} + \boldsymbol{D}$.

**Step 2:** We apply $\mathcal{T}(\cdot)$ to the matrices $\boldsymbol{B} + \boldsymbol{D}$ and $\boldsymbol{B}$ in Eq. (61). Since the transformation is invariant, we have:

$$\mathcal{T}(\boldsymbol{B} + \boldsymbol{D})\hat{\boldsymbol{M}} = \mathcal{T}(\boldsymbol{B}) \tag{62}$$

Thus:

$$P_k P_{k-1} \cdots P_1(\boldsymbol{B} + \boldsymbol{D})P_1 P_2 \cdots P_{k-1} P_k \hat{\boldsymbol{M}} = P_k P_{k-1} \cdots P_1(\boldsymbol{B})P_1 P_2 \cdots P_{k-1} P_k \tag{63}$$

We can easily see that if $\hat{\boldsymbol{M}}$ is a solution to this equation, then $\mathcal{T}(\hat{\boldsymbol{M}}) = P_k P_{k-1} \cdots P_1 \hat{\boldsymbol{M}} P_1 P_2 \cdots P_{k-1} P_k$ is also a solution, since $P_i^2 = I, i = 1, \cdots, k$, where $I$ is the identity matrix. In addition, because $\boldsymbol{B} + \boldsymbol{D}$ is invertible (as shown in Appendix F.4), this solution is unique. Therefore:

$$\mathcal{T}(\hat{\boldsymbol{M}}) = \hat{\boldsymbol{M}} \tag{64}$$

This shows that the transformation $\mathcal{T}(\cdot)$ also maps $\hat{\boldsymbol{M}}$ to itself.

**Conclusion:** We have shown that, under the transformation $\mathcal{T}(\cdot)$, the affected rows of $\hat{\boldsymbol{M}}$ are permutations of each other. Note that only the rows with the same order will be permuted to each other because $\mathcal{T}(\cdot)$ is derived from the permutation of the power set of $N$, so the order of the rows is preserved.

For any two subsets $T, T' \subseteq N$ of the same order, we can construct a permutation of indices from $T$ to $T'$ that maps $\hat{\boldsymbol{m}}_T$ to $\hat{\boldsymbol{m}}_{T'}$. Therefore, $\hat{\boldsymbol{m}}_T$ is a permutation of $\hat{\boldsymbol{m}}_{T'}$. $\qquad\square$

| Output function $v(\cdot)$ | $v(\boldsymbol{x}) = \log \frac{p(y^{\text{truth}}|\boldsymbol{x})}{1-p(y^{\text{truth}}|\boldsymbol{x})}$ |
|---|---|
| Threshold $\tau$ | $\tau = 0.03\,\mathbb{E}_{\boldsymbol{x}}[|v(\boldsymbol{x}) - v(\boldsymbol{x}_\emptyset)|]$ |
| Baseline value $\boldsymbol{b}$ | Image data: using the zero baseline on the feature map after ReLU |
| | Text data: using the [MASK] token |
| | Point cloud data: using the cluster center of each point cluster |

Table 1: Mathematical setting of hyper-parameters for interactions.

### F.7   Proof of Theorem 5

*Proof.* From Eq. (10), when there is no noise (*i.e.*, $\sigma = 0$), it is obvious that $\hat{\boldsymbol{w}} = (\boldsymbol{J}^\top \boldsymbol{J})^{-1}\boldsymbol{J}^\top \boldsymbol{J}\boldsymbol{w}^* = \boldsymbol{w}^*$, which means that the optimal weights $\hat{\boldsymbol{w}}$ are the same as the true weights $\boldsymbol{w}^*$. So $\forall\, \emptyset \neq T \subseteq N,\ \hat{w}_T = w_T^*$. $\hspace{1cm}\square$

## G   Experimental details

### G.1   Models and datasets

We trained various DNNs on different datasets. Specifically, for image data, we trained VGG-11 on the MNIST dataset (Creative Commons Attribution-Share Alike 3.0 license), VGG-11/VGG-16 on the CIFAR-10 dataset (MIT license), AlexNet/VGG-16 on the CUB-200-2011 dataset (license unknown), and VGG-16 on the Tiny ImageNet dataset (license unknown). For natural language data, we trained BERT-Tiny and BERT-Medium on the SST-2 dataset (license unknown). For point cloud data, we trained DGCNN on the ShapeNet dataset (Custom (non-commerical) license).

For the CUB-200-2011 dataset, we cropped the images to remove the background regions, using the bounding box provided by the dataset. These cropped images were resized to 224×224 and fed into the DNN. For the Tiny ImageNet dataset, due to the computational cost, we selected 50 classes from the total 200 classes at equal intervals (*i.e.*, the 4th, 8th,..., 196th, 200th classes). All these images were resized to 224×224. For the MNIST dataset, all images were resized to 32×32 for classification. To better demonstrate that the learning of higher-order interactions in the second phase was closely related to overfitting, we added a small ratio of label noise to the MNIST dataset, the CIFAR-10 dataset, and the CUB-200-2011 dataset to boost the significance of over-fitting of the DNNs. Specifically, we randomly selected 1% training samples in the MNIST dataset and the CIFAR-10 dataset, and randomly reset their labels. We randomly selected 5% training samples in the CUB-200-2011 dataset and randomly reset their labels.

### G.2   Training settings

We trained all DNNs using the SGD optimizer with a learning rate of 0.01 and a momentum of 0.9. No learning rate decay was used. We trained VGG models, AlexNet models, and BERT models for 256 epochs, and trained the DGCNN model for 512 epochs. The batchsize was set to 128 for all DNNs on all datasets.

### G.3   Details on computing interactions

First, we provide a summary of the mathematical settings of the hyper-parameters for interactions in Table 1, including the scalar output function of the DNN $v(\cdot)$, the baseline value $\boldsymbol{b}$ for masking, and the threshold $\tau$. These settings are uniformly applied to all DNNs. More detailed settings for different datasets can be found below.

**Image data.** For image data, we considered image patches as input variables to the DNN. To generate a masked sample $\boldsymbol{x}_S$, we followed [41] to mask the patch on the intermediate-layer feature map corresponding to each image patch in the set $N \setminus S$. Specifically, we considered the feature map after the second ReLU layer for VGG-11/VGG-16 and the feature map after the first ReLU layer for AlexNet. For the VGG models and the AlexNet model, we uniformly partitioned the feature map into 8×8 patches, randomly selected 10 patches from the central 6×6 region (*i.e.*, we did not select patches that were on the edges), and considered each of the 10 patches as an input variable in the set $N$ to calculate interactions. We considered each of the 10 patches as an input variable in the set $N$ to calculate interactions. We used a zero baseline value to mask the input variables in the set $N \setminus S$ to obtain the masked sample $\boldsymbol{x}_S$.

**Natural language data.** We considered the input tokens as input variables for each input sentence. Specifically, we randomly selected 10 words that are meaningful (*i.e.*, not including stopwords, special characters, and punctuations) as input variables in the set $N$ to calculate interactions. We used the "mask" token with the token id 103 to mask the tokens in the set $N \setminus S$ to obtain the masked sample $\boldsymbol{x}_S$.

**Point cloud data.** We clustered all the points into 30 clusters using K-means clustering, and randomly selected 10 clusters as the input variables in the set $N$ to calculate interactions. We used the average coordinate of the points in each cluster to mask the corresponding cluster in $N \setminus S$ and obtained the masked sample $\boldsymbol{x}_S$.

For all DNNs and datasets, we randomly selected 50 samples from the testing set to compute interactions, and averaged the interaction strength of the $k$-th order on each sample to obtain $I_{\text{real}}^{(k)}$.

### G.4 Compute resources

All DNNs can be trained within 12 hours on a single NVIDIA GeForce RTX 3090 GPU (with 24G GPU memory). Computing all interactions on a single input sample usually takes 35-40 seconds, which is acceptable in real applications.

## H Potential limitations of the theoretical proof

In this study, we have assumed that during the training process, the noise on the parameters gradually decreased ($\sigma^2$ gradually became smaller). Although experiments in Figure 4 and Figure 8 have verified that the theoretical distribution of interaction strength can well match the real distribution by using a set of decreasing $\sigma^2$ values, it is not exactly clear how the value of $\sigma^2$ is related to the training process. The value of $\sigma^2$ probably does not decrease linearly along with the training epochs/iterations, which needs more precise formulations.

## I More discussions about the two-phase dynamics

### I.1 Does the model re-learn the initial interactions during the second phase?

Our theory does **not** claim that in the second phase, a DNN will not re-encode an interaction that is removed in the first phase. Instead, Theorem 4 and Proposition 1 collectively indicate the possibility of a DNN gradually re-encoding a few higher-order interactions in the second phase along with the decrease of the parameter noise.

The key point to this question is that the massive interactions in a fully initialized DNN are all chaotic and meaningless patterns caused by randomly initialized network parameters. Therefore, the crux of the matter is not whether the DNN re-learns the initially removed interactions, but the fact that the DNN mainly removes *chaotic and meaningless initial interactions* in the first phase, and learns *potential target interactions* in the second phase. In this way, although a few interactions may be re-encoded later in the second phase, we do not consider this as a problem with the training of a DNN.

### I.2 About extending the theoretical analysis to specific network architectures

Our current analysis is agnostic to the network architecture, and aims to explain the common two-phase dynamics of interactions that is *shared by different network architectures for various tasks*. Fig. 2 and Fig. 5 demonstrate this shared two-phase dynamics.

On the other hand, although DNNs with different architectures all exhibit the two-phase dynamics of interactions, the length of the two phases and the finally converged state of the DNN are influenced by the network architecture and can slightly vary among different architectures. Eq. (10) shows that our current formulation is to use the finally converged state of a DNN to accurately predict the DNN's learning dynamics of interactions. Therefore, the learning dynamics predicted by our theory also exhibits slight differences among different DNN architectures and datasets accordingly, but it still matches well with the empirical dynamics of interactions. To this end, studying how the network architecture affects the finally converged state of a DNN may be a good future direction.

# J More experimental results

## J.1 More results for the two-phase phenomenon

In this subsection, we show the two-phase dynamics of learning interactions on more DNNs and datasets. See Figure 5 and Figure 6 for details.

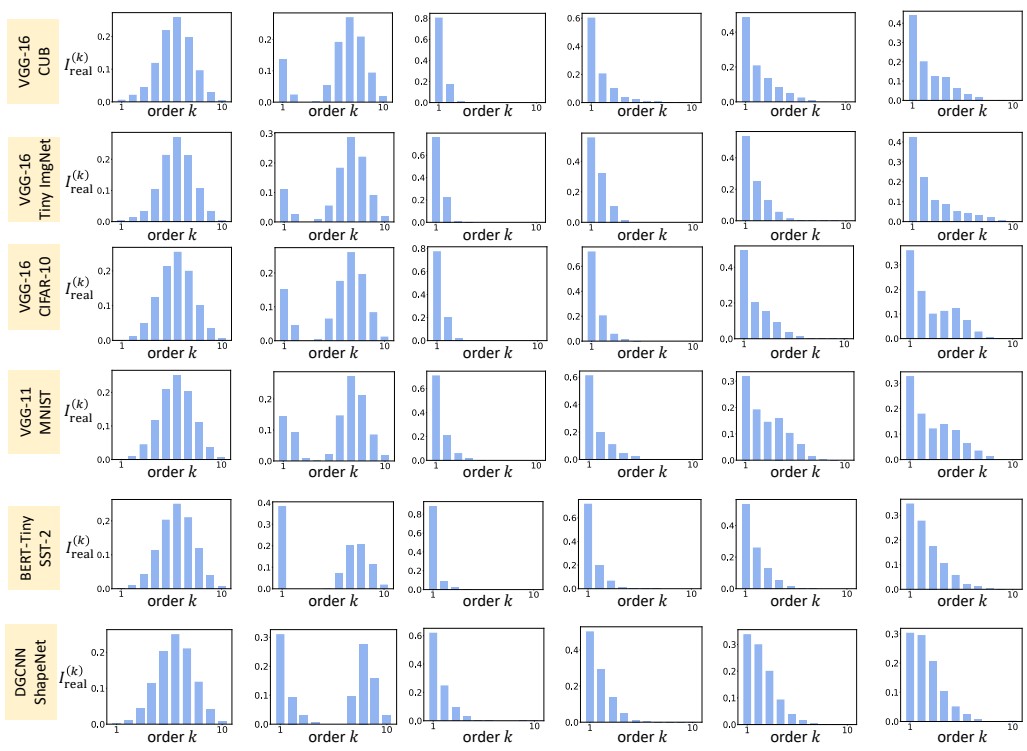

Figure 5: The distribution of interaction strength $I_{\text{real}}^{(k)}$ over different orders $k$. Each row shows the change of the distribution during the training process. Experiments showed that the two-phase phenomenon widely existed on different DNNs trained on various datasets.

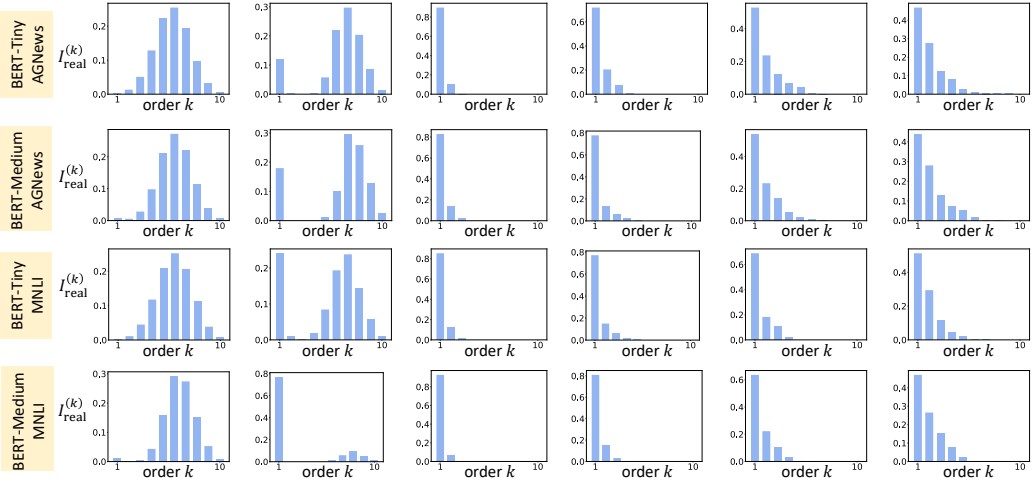

Figure 6: Demonstration of the two-phase dynamics of interactions on more textual datasets.

## J.2 More details for the alignment between the two phases and the loss gap

Besides the loss gap, in Figure 7, we also show the training loss and the testing loss separately. In fact, instead of considering underfitting (or learning useful features) and overfitting (or learning overfitted features) as two separate processes, the DNN simultaneously learns both useful features and overfitted features during training. The learning of useful features decreases the training loss and the testing loss, which alleviates underfitting. Meanwhile, the learning of overfitted features gradually increases the loss gap.

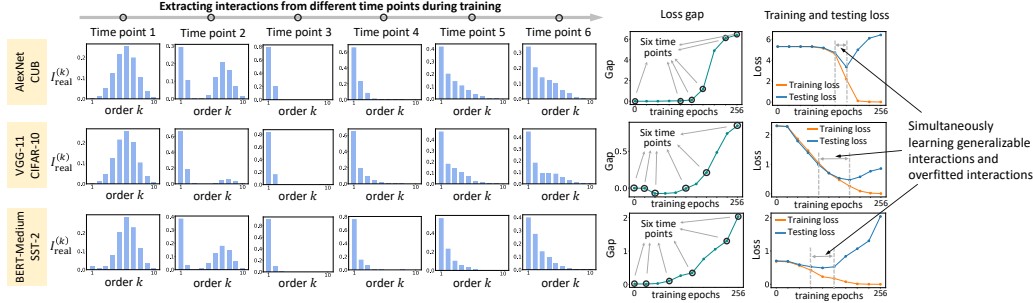

Figure 7: Demonstration of the training loss and the testing loss (the last column) in addition to the two-phase dynamics of interactions (1st column to 6th column) and the loss gap (7th column).

## J.3 More results for the experimental verification of our theory

In this subsection, we show results of using the theoretical distribution of interaction strength $I_{\text{theo}}^{(k)}$ to match the real distribution of interaction strength $I_{\text{real}}^{(k)}$ on more DNNs and datasets, as shown in Figure 8.

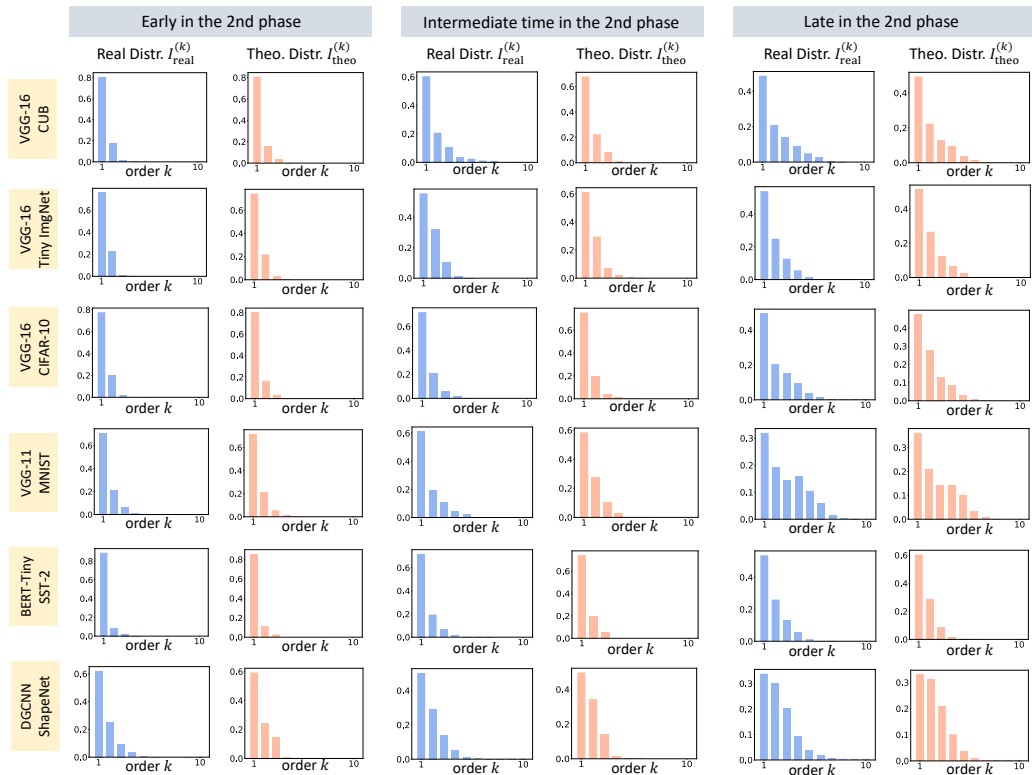

Figure 8: Comparison between the theoretical distribution of interaction strength $I_{\text{theo}}^{(k)}$ and the real distribution of interaction strength $I_{\text{real}}^{(k)}$ in the second phase on more DNNs and datasets.

## J.4 Using the theoretical distribution $I_{\text{theo}}^{(k)}$ to predict the real distribution of AND interactions

In this subsection, we show results of using the theoretical distribution of interaction strength $I_{\text{theo}}^{(k)}$ to match the real distribution of AND interactions (rather than the AND-OR interactions), as shown in Figure 9.

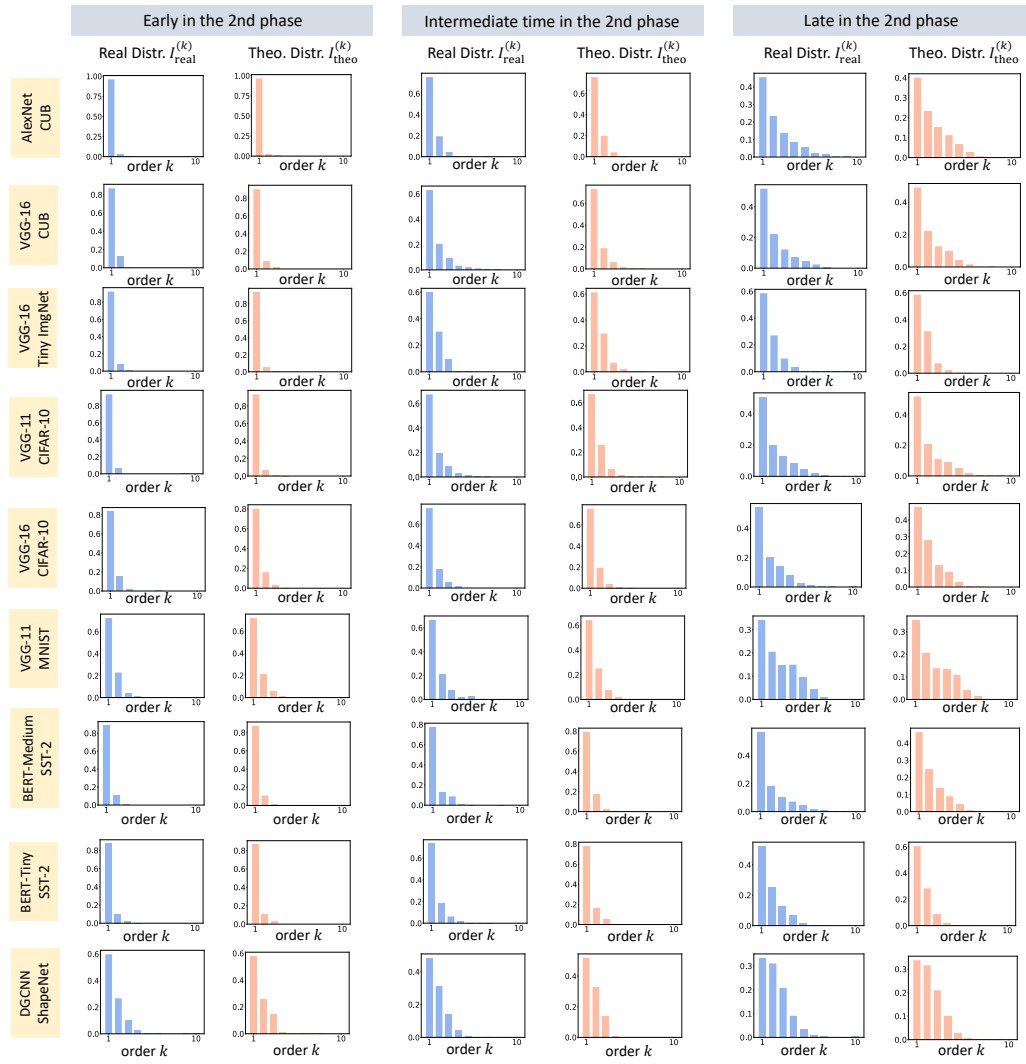

Figure 9: Comparison between the theoretical distribution of interaction strength $I_{\text{theo}}^{(k)}$ and the real distribution of interaction strength of AND interactions.

