# OpenReview forum: "Towards the Dynamics of a DNN Learning Symbolic Interactions"
_NeurIPS.cc/2024/Conference — NeurIPS 2024 poster_

### Official Review · Reviewer_xnzG · 2024-06-19

**Soundness:** 3
**Presentation:** 3
**Contribution:** 3
**Rating:** 8
**Confidence:** 3

**Summary:**

This paper studies the training dynamics (underfitting to overfitting) of deep neural networks via the perspective of symbolic interactions. They formulate the learning of interactions as a linear regression problem on a set of interaction triggering functions. They show the two-stage dynamics. In the first stage, neural networks first remove initial interactions and learn low-order interactions. In the second stage, neural networks learn increasingly more complicated interactions, leading to overfitting. Through empirical experiments, they show the story is valid for various architectures.

**Strengths:**

* The paper is well written, clear in motivation. Figures are pleasure to read.
* Statements are justified with math theorems.
* The link between underfitting-overfitting vs the distribution of I_k is novel.
* Testing extensively on various architectures

**Weaknesses:**

* The contribution is not fully clear.  Since this work heavily relies on [26][27][45], it should clearly highlight what's new and what's known in pervious works.

**Questions:**

* Line 21, "the three conditions". what are these three conditions?
* Line 21, "for" -> "For"
* Line 100, Eq. (2), can you explain more where does the factor (-1)^{|S|-|T|} comes from? This sign seem to be canceling terms out?
* In Figure 2, it would be nice to also include training loss, not just train-test gap. Is there a time point when the interactions are too simple such that increasing order actually helps alleviate underfitting but still not overfitting?
* How does the initial I_k distribution change with different initialization scales?

**Limitations:**

The authors adequately addressed the limitations.

---

> ### Author Rebuttal · Authors · 2024-08-07
>
> Thank you for your comments. We are glad to answer all your questions. **If you have new questions, please let us know as soon as possible.**
>
> **Q1: Ask for further clarification on the paper's contribution.**
>
> > The contribution is not fully clear. Since this work heavily relies on [26] [27] [45], it should clearly highlight what's new and what's known in previous works.
>
> A: Thank you. We will follow your suggestion to further clarify a list of substantial contributions over previous works [26, 27, 45].
>
> (1) [27] is a theoretical foundation for our paper. As introduced in Lines 20-22 and Appendix B, [27] just proves that under three conditions, the a DNN's decision-making logic can be faithfully explained by a few symbolic interactions. Based on [27], we aim to further prove the two-phase learning dynamics of interactions.
>
> (2) We just use the re-formulation of the linear representation of interactions in [26] as a mathematical tool to prove the learning dynamics of interactions. In comparison, [26] just aims to discover the interaction bottleneck of Bayesian neural networks, which is fully different from our task of proving the learning dynamics of interactions on all types of DNNs.
>
> (3) Our proof of the learning dynamics does not have any direct connection with [45]. However, by combining the proven dynamics and [45]’s finding that "high-order interactions have weaker generalization power," we will obtain the following significant conclusion: our derived dynamics of interactions also explain the dynamics of the generalization power of a DNN. This has been introduced in Lines 170-190.
>
> ---
>
> **Q2:** "Line 21, 'the three conditions'. what are these three conditions?  "
>
> A: We have provided the three conditions in Appendix B. Please see Appendix B for details. In addition, we will revise Line 21 to make the reference to Appendix B clearer.
>
> Specifically, the three conditions can be intuitively understood as requiring a DNN to generate relatively smooth inference scores on masked samples, and [27] shows that the three conditions are quite common and can be satisfied by DNNs trained for many real applications.
>
> Note that the proof of interaction dynamics in this paper does not depend on the three conditions. The three conditions are just used to prove the sparsity of interactions.
>
> ---
>
> **Q3:** "Line 21, 'for' -> 'For'. "
>
> A: Thank you. We will correct this typo.
>
> ---
>
> **Q4: Ask about the computation of interactions.**
>
> > Line 100, Eq. (2), can you explain more where does the factor (-1)^{|S|-|T|} comes from? This sign seem to be canceling terms out?
>
> A: A good question. In fact, it is proven in Appendix E.2 that the coefficient $(-1)^{|S|-|T|}$ in Eq. (2) is the **unique** coefficient to ensure that the interaction satisfies the **universal matching property**. The universal matching property (in Line 105) means that no matter how we randomly mask an input sample ${x}$, the network output on the masked sample $x_S$ can always be accurately mimicked by the sum of interaction effects within $S$.  An extension of this property for AND-OR interactions is also mentioned in Theorem 2.
>
> Nevertheless, we will clarify how the coefficient $(-1)^{|S|-|T|}$ is derived in the paragraph following Eq. (2).
>
> ---
>
> **Q5: Ask for including the training loss in Fig. 2**
>
> > In Figure 2, it would be nice to also include training loss, not just train-test gap. Is there a time point when the interactions are too simple such that increasing order actually helps alleviate underfitting but still not overfitting?
>
> A: We have followed your suggestion to show the training loss in Fig. 2. Please see the last column of Fig. 2 in the *response pdf file* for details. In fact, instead of considering underfitting (or learning useful features) and overfitting (or learning overfitted features) as two separate processes, the DNN simultaneously learns both useful features and overfitted features during training. The learning of useful features decreases the training loss and the testing loss, which alleviates underfitting. Meanwhile, the learning of overfitted features gradually increases the loss gap.
>
> ---
>
> **Q6:** "How does the initial I_k distribution change with different initialization scales?"
>
> A: A good question. A short answer is that initializing a DNN with different scales only uniformly rescales the magnitudes of all interactions $\forall S\subseteq N, I^{new}(S|x)=\lambda\ I^{old}(S|x)$ by a constant $\lambda$, but does not change the distribution $I^{(k)}$ of different interactions.
>
> **Theoretical analysis.** Let us take the output of a ReLU network corresponding to the target category before the softmax layer as the function $v$. If all randomly initialized parameters (the biases are typically initialized to zero) in the ReLU network are scaled with a constant $c$, $\theta^{new}=c\ \theta^{old}$, then the value of all interactions will be scaled to $\lambda=c^L$ times of their original values, where $L$ is the number of layers of the network. This is because the scaling of all initial parameters will cause the network output on any input to be scaled to $\lambda=c^L$ times of the original value, $v^{new}(x)=c^L\ v^{old}(x)$. Then, scaling the output $v(x)$ by $\lambda$ will also scale the interaction $I(S|x)$ by $\lambda$, according to Eq. (2). However, since all interactions are multiplied by the same constant $\lambda=c^L$, the *normalized distribution* $I^{(k)}$ across different orders (defined in Line 159) remains the same.
>
> We have also conducted **new experiments** to verify that initial $I^{(k)}$ distribution does not change with different initialization scales. We tested VGG-16 on the CIFAR-10 dataset. Fig. 3 in the *response pdf file* shows that when we scaled the initialized parameters with a constant $c=0.5$, $c=1.5$, and $c=2$, only the magnitude of interactions changed, but the distribution of interactions remained the same.

---

> > ### Comment · Reviewer_xnzG · 2024-08-11
> >
> > I want to thank the author for addressing my concerns. I'm more convinced now this is an important paper for its novel concepts. I'll raise my score to 8.

---

> > > ### Author Response · Authors · 2024-08-12
> > >
> > > Thank you very much for your appreciation. We will continue to enhance the paper according to the discussion with all reviewers. We hope this paper provides deep insights into the two-phase dynamics of interactions and its tight connection to the generalization power of DNNs.

---

### Official Review · Reviewer_bep2 · 2024-07-12

**Soundness:** 3
**Presentation:** 3
**Contribution:** 2
**Rating:** 7
**Confidence:** 3

**Summary:**

This study investigates the two-phase dynamics of DNNs learning interactions during training, demonstrating that DNNs initially focus on simpler, low-order interactions and progressively transition to more complex, high-order interactions. The learning process is reformulated as a linear regression problem, which facilitates the analysis of how DNNs manage interaction learning under parameter noise.

**Strengths:**

- The research backs its theoretical claims with sufficient experimental evidence. This thorough analysis strengthens the credibility of the study's conclusions.
- The study explores the two-phase dynamics of DNNs, clarifying how networks transition from simple to complex interactions. This clarifies the mechanisms underlying neural network generalization and susceptibility to overfitting.

**Weaknesses:**

- The experiments on textual is limited compared to those on vision tasks.

**Questions:**

- How is it ensured that the model does not re-learn the initial interactions during the second phase? Please justify.
- Please justify the disparity in VGG-16 on CIFAR-10 and VGG-11 MNIST based on Figure 5. It seems their distribution is a bit different from the others in the last time point.

**Limitations:**

- The experiments on textual data are limited. Expanding the analysis to include a broader range of datasets could provide a more comprehensive understanding of the DNN's interactions.

---

> ### Author Rebuttal · Authors · 2024-08-07
>
> Thank you for your comments. We are glad to answer all your questions. **If you have new questions, please let us know as soon as possible, so that we can try our best to answer any further questions in the discussion period.**
>
> **Q1: Ask for experiments on more textual datasets.**
>
> > The experiments on textual data are limited. Expanding the analysis to include a broader range of datasets could provide a more comprehensive understanding of the DNN's interactions.
>
> A: Thank you. We have followed your suggestion to conduct **new experiments** on more natural language datasets, including the AG News dataset [cite1] for news classification, and the MNLI dataset [cite2] for natural language inference. We train the BERT-Tiny model and the BERT-Medium model on these datasets. Fig. 1 in the *response pdf file* shows that on all these models and datasets, the dynamics of interactions over different orders during the training process all exhibit the two-phase phenomenon.
>
> [cite1] Zhang et al. Character-level Convolutional Networks for Text Classification. NeurIPS 2015.
>
> [cite2] Williams et al. A Broad-Coverage Challenge Corpus for Sentence Understanding through Inference. NAACL 2018.
>
> ---
>
> **Q2:** "How is it ensured that the model does not re-learn the initial interactions during the second phase? Please justify."
>
> A: A good question. Our theory does **not** claim that in the second phase, a DNN will not re-encode an interaction that is removed in the first phase. Instead, Theorem 4 and Proposition 1 both indicate the possibility of a DNN gradually re-encoding a few higher-order interactions in the second phase along with the decrease of the parameter noise.
>
> Essentially, we think the key point to this question is that the massive interactions in a fully initialized DNN are all chaotic and meaningless patterns caused by randomly initialized network parameters. Therefore, the crux of the matter is not whether the DNN re-learns the initially removed interactions,  but the fact that the DNN mainly removes *chaotic and meaningless initial interactions* in the first phase, and learns *target interactions* in the second phase. In this way, although a few interactions may be re-encoded later in the second phase, we do not consider this as a problem with the training of a DNN.
>
> ---
>
> **Q3: Ask about slightly different distributions of the finally learned interactions in Figure 5.**
>
> > Please justify the disparity in VGG-16 on CIFAR-10 and VGG-11 MNIST based on Figure 5. It seems their distribution is a bit different from the others in the last time point.
>
> A: Thank you for your comments. In fact, our conclusion that "DNNs with different architectures all share the same two-phase dynamics" does **not** mean that different DNNs all encode exactly the same distributions of interactions. Instead, the distributions of interactions encoded by different DNNs may be slightly different, but their dynamics consistently exhibit two phases.
>
> It is because Eq. (10) in our paper has shown that the distribution of interactions is determined by the state of the finally converged DNN after the last epoch, and feature representations of the finally converged DNN are affected by the network architecture and the dataset. Thus, the learning dynamics predicted by our theory also accordingly exhibit such slight differences among different DNNs and datasets.
>
> Besides, experimental results in Fig. 4 and Fig. 6 also show that our theory can well predict the slight difference of the interaction distributions between different DNNs throughout the entire training process. I.e., our theory predicts different interaction distributions for different DNNs and datasets.

---

> > ### Comment · Reviewer_bep2 · 2024-08-09
> >
> > The authors have clarified my questions. I would recommend that the authors include the explanation given for "Q2: re-learning initial interactions" in the final version of the paper if accepted. I have decided to raise my score.

---

> > > ### Author Response · Authors · 2024-08-10
> > >
> > > Thank you very much. We will follow your suggestion to incorporate the explanation for "Q2: re-learning initial interactions" into the paper if the paper is accepted.

---

### Official Review · Reviewer_iitK · 2024-07-13

**Soundness:** 2
**Presentation:** 1
**Contribution:** 2
**Rating:** 5
**Confidence:** 2

**Summary:**

The paper investigates the two-phase dynamics of interactions during training by reformulating the learning of interactions as a linear regression problem. The authors provide an analytic solution to the minimization problem and use this solution to explain the two-phase dynamics of interactions.

**Strengths:**

The paper provides **a theoretical explanation for the two-phase dynamics of interactions**, which appears to be a novel contribution to the literature.

**Weaknesses:**

* **Presentation of the paper is difficult to follow**, especially for someone like me who is not familliar with the literature. It would benefit from improved clarity and readability, particularly in summarizing relevant literature and explaining the mathematical setting.
* The paper only characterizes minimizers, which are the ending points of learning and **do not consider the exact training dynamics**.

**Questions:**

* Do you think **it is possible to analyze the exact dynamics** of interactions by considering specific neural network architectures and data models? This direction may enhance the results of the paper.
* Are there any **practical implications** of the theoretical findings?

---

> ### Author Rebuttal · Authors · 2024-08-07
>
> Thank you for your great efforts on the review. We will answer all your questions. **If you have new questions, please let us know as soon as possible. Thank you.**
>
> **Q1:** "Presentation of the paper is difficult to follow ... particularly in summarizing relevant literature and explaining the mathematical setting."
>
> A: Thank you. We will follow your suggestions to improve the presentation as follows.
>
> $\bullet$ First, we will revise the related work section to add a more thorough summarization of relevant literature on interactions. The theory system of interactions has been surveyed in both [27] and the related work section in this paper, which consists of more than 15 papers published in top-tier conferences/journals (T-PAMI, NeurIPS, ICML, ICLR, CVPR, etc.) since 2021. The theory system of interactions explains DNNs from different perspectives. E.g., (1) proving that the decision-making logic of a DNN on a sample can be explained by a small number of symbolic interactions [21,23,27]; (2) proving that the interactions well explain the hidden factors that determine the generalization power, robustness, and adversarial transferability of a DNN [24,37,41,45]; (3) proving that internal mechanisms of many empirical deep learning methods can be reformulated using interactions (e.g., finding that the common essence of 14 attribution methods is the reallocation of interactions [8]).
>
> $\bullet$ Second, regarding the mathematical setting of interactions, we only need to set hyper-parameters including the scalar output function of the DNN $v(\cdot)$, the baseline value $\boldsymbol{b}$ for masking, and the threshold $\tau$. Detailed settings of $v(\cdot)$, $\boldsymbol{b}$, and $\tau$ have been introduced in [Line 85 and Footnote 3], [Footnote 4 and Appendix F.3], and Line 157, respectively. These settings are uniformly applied to all DNNs. Alternatively, you may also see Table 1 in the *response pdf file* for these settings. Nevertheless, we will also clarify these in the main paper.
>
> ---
>
> **Q2: Does the "characterization of minimizers" represent the end point of learning or the intermediate state in the training dynamics?**
>
> > The paper only characterizes minimizers, which are the ending points of learning and do not consider the exact training dynamics.
>
> A: Thank you. In fact, the characterization of the minimizer (i.e., the optimal solution to Eq. (9)) **does not** represent the state of the end of learning, but represents the *intermediate state* of interactions after *a certain epoch in the training dynamics*. This is because as mentioned in Lines 239-247, we formulate the training process as a process of gradually reducing the noise on the DNN's parameters, and the minimizer $\hat{\boldsymbol{w}}$ to Eq. (9) represents the optimal interaction state when the training of the DNN is subject to unavoidable *parameter noises*. In this way, the minimizer $\hat{\boldsymbol{w}}$ computed under different noise levels accurately predicts the exact dynamics of interactions (please see Fig. 4 and Fig. 6), because the parameter noise is the key factor that controls the training dynamics in the second phase.
>
> Nevertheless, we will further clarify this in Line 239-247. We hope this answer helps address your concern, and please let us know if you have new concerns.
>
> ---
>
> **Q3: Ask about extending the theoretical analysis to specific network architecture.**
>
> > Do you think it is possible to analyze the exact dynamics of interactions by considering specific neural network architectures and data models? This direction may enhance the results of the paper.
>
> A: A good question. First, we would like to clarify that the analysis is agnostic to the network architecture. To be precise, a recent work [27] has shown that except for cases where an extremely poor architecture fully damages the feature representation of a DNN, architectures for most typical models (including both SOTA models and sub-optimal models ranging from the MLP, the LeNet to transformer-based LLMs) are no longer the key factor that leads to the emergence of sparse interactions. Instead, the property of stable inference on masked samples is the direct cause for the emergence of sparse interactions. Similarly, *the two-phase dynamics of interactions in this paper is shared by different network architectures for various tasks*. Please see Fig. 2 and Fig. 5 for this two-phase dynamics.
>
> On the other hand, although DNNs with different architectures all exhibit the two-phase dynamics of interactions, the length of the two phases and the finally converged state of the DNN are also influenced by the network architecture. Eq. (10) shows that our current progress is to use the finally converged state of a DNN to accurately predict the DNN's learning dynamics of interactions. In this way, how the network architecture affects the finally converged state of a DNN is also a good future direction.
>
> ---
>
> **Q4:** "Are there any practical implications of the theoretical findings?"
>
> A: A good question. A theoretical understanding of the two-phase dynamics of interactions provides a new perspective to monitor the overfitting level of the DNN on different training samples throughout the entire training process. Previous metric (e.g., the loss gap between the training set and testing set) only measures the overfitting level of the DNN over the entire dataset. In comparison, the discovered two-phase dynamics enable us to evaluate the overfitting level of each specific training sample, making overfitting no longer a problem w.r.t. the entire dataset. We can track the change of the interaction complexity for each training sample, and take the time point when high-order interactions increase as a sign of overfitting. In this way, the two-phase dynamics of interactions may help people remove overfitted samples from training and guide the early stopping of a few "hard samples." We will clarify this in the paper.

---

> > ### Comment · Reviewer_iitK · 2024-08-11
> >
> > Thank you for your detailed response. The response adequately addressed my concerns. After reviewing the discussion between the authors and reviewers and re-reading the draft, I gained a better understanding of the work. As a result, I have decided to increase my score. I hope the authors will further enhance the paper by incorporating the more detailed background and discussion points mentioned in their response. I believe these improvements will make the paper more accessible to readers who are less familiar with this literature.

---

> > > ### Author Response · Authors · 2024-08-12
> > >
> > > Thank you for very much. We will follow your suggestion to provide a more thorough review of the background and incorporate the discussion points in the next version of the paper if accepted.

---

### Author Rebuttal · Authors · 2024-08-07

We would like to thank all reviewers for the constructive comments and questions. We have carefully considered all your comments and answered all the questions, and will revise the paper to clarify all your concerns. In addition, we have followed your suggestions to conduct **new experiments**, please see the **response pdf file** for results.

**Please let us know if you have further questions, so that we can get back to you as soon as possible.**

---

### Decision · Program_Chairs · 2024-09-25

**Decision:**

Accept (poster)

**Comment:**

The paper gives a theoretical study of how the dynamics of learned symbolic interactions in DNNs evolve. The paper gives theoretical and empirical evidence for a two-phase learning dynamic for DNNs on data with sparse complex interactions.

While there has been some concern over the clarity of the paper and the overall message, reviewers appreciated the strength of the theoretical claims, backed with empirical evidence, and agree that this paper should be accepted. I therefore recommend accepting the paper.